



# Multi-fidelity model assessment of climate change impacts on river water temperatures, thermal extremes and potential effects on cold water fish in Switzerland

Love Råman Vinnå[1], Vidushi Bigler[2], Oliver S. Schilling[3,4], Jannis Epting[*1]

[1]*Applied and Environmental Geology, Hydrogeology, Department of Environmental Sciences, University of Basel, CH-4056 Basel, Switzerland*

[2]*Bern University of Applied Sciences, Engineering and Computer Science (BFH-TI), Institute for Optimization and Data Analysis (IODA), 2501 Biel, Switzerland*

[3]*Hydrogeology, Department of Environmental Sciences, University of Basel, CH-4056 Basel, Switzerland*

[4]*Department Water Resources and Drinking Water, Eawag - Swiss Federal Institute of Aquatic Science and Technology, CH-8600 Dübendorf, Switzerland*

[*]Corresponding author: jannis.epting@unibas.ch

OrcIDs:

Love Råman Vinnå: https://orcid.org/0000-0002-9108-8057

Jannis Epting: https://orcid.org/0000-0001-9578-5557

Oliver S. Schilling: https://orcid.org/0000-0003-3840-7087





# 1 Abstract

River water temperature is a key factor for water quality, aquatic life, and human use. Under
climate change, inland water temperatures have increased and are expected to do so further,
increasing the pressure on aquatic life and reducing the potential for human use. Here, future
river water temperatures are projected for Switzerland based on a multi-fidelity modelling
approach. We use 2 different, semi-empirical surface water temperature models, 22 coupled
and downscaled general circulation- to regional climate models, future projections of river
discharge from 4 hydrological models and 3 climate change scenarios (RCP2.6, 4.5, and 8.5).
By grouping stream sections, catchments and spring-fed water courses under representative
thermal regimes, and by employing hierarchical cluster-based thermal pattern recognition, an
optimal model and model configuration was selected, model performance optimized and
climate change impact assessment on river water temperatures improved.

Results show that, until the end of the 21$^{st}$ century, average river water temperatures in
Switzerland will likely increase by 3.1±0.7 °C (or 0.36±0.1 °C per decade) under RCP8.5,
while under RCP2.6 the temperature increase may remain at 0.9±0.3 °C (0.12±0.1 °C per
decade). Under RCP8.5, temperatures of rivers classified as being in the Alpine thermal regime
will increase the most, that is, by 3.5±0.5 °C, followed by rivers of the Downstream Lake
regime, which will increase 3.4±0.5 °C.

A general decrease of river discharge in summer (-10 to -40 %) and increase in winter (+10 to
+30%), combined with a further increase in average near-surface air temperatures (0.5 °C per
decade), bears the potential to not only result in overall warmer rivers, but also in prolonged
periods of extreme summer river water temperatures. This dramatically increases the thermal
stress potential for temperature sensitive aquatic species such as the brown trout in rivers where
such periods occur already, but also rivers in where this previously was not a problem. By
providing information of future water temperatures, the results of this study can guide
managements climate mitigation efforts.



## 1 Introduction

River water temperature is a key factor in the regulation of physical and biogeochemical processes in aquatic systems, affecting water quality, aquatic life and the potential for human water use. Globally, climate change has already increased, and is expected to further increase, river water temperatures (Van Vliet et al., 2011; 2013). Without climate protection, it is estimated that, globally, 36% of fish species will see their future habitats exposed to climate extremes, with changes in water temperatures being deemed more critical than the change in water availability (Barbarossa et al., 2021). The amount of river warming, especially during heat waves and droughts, is however not only a function of near-surface air temperatures, but also of river discharge, river-groundwater interactions, and human activities such as channelization, damming, water use for cooling purposes, or sewage and storm water runoff all affecting water quality (Ficklin et al., 2023; Van Vliet et al., 2023).

In Switzerland, the water tower of Europe, the effects of a changing climate have already influenced both river temperatures (Hari & Güttinger, 2004) and river discharge (Birsan et al., 2005). According to the latest regional climate projections (CH2018, 2018) the change is likely to continue to affect Swiss waterbodies in the future (FOEN, 2021). Past water temperature trends in Switzerland from 1979 to 2018 amounted to an increase of 0.33 °C per decade on average, alongside a near-surface air temperature increase of 0.46 °C per decade (Michel et al., 2020). Using a limited subset of federally monitored Swiss catchments (~10%) and a high emission climate scenario (RCP8.5), it was projected that water temperatures may continue to increase by 3.5 °C until the end of the 21st century (Michel et al., 2022). Being a higher elevation country (mean elevation 1'350 mASL), most rivers in Switzerland are populated by the brown trout (*salmo trutta fario),* a cold-water fish (Brodersen et al., 2023). All fish species have specific temperature limits within which optimal conditions for growth, health, reproduction, or life, exist. For the brown trout, which is a particularly temperature sensitive fish species, warmer water temperatures of around 13°C pose a threat for egg survival, 15°C strongly increases their receptivity for parasites related illnesses, and prolonged exposure to 25°C can lead to death (Strepparava et al., 2018; Wehrly et al., 2007; Chilmonczyk et al., 2002; Elliott, 1994). A prime example of a water temperature related threat is the elevation (i.e., water temperature) dependent proliferative kidney disease (PKD), a parasite-caused illness in brown trout which is increasingly wide-spread in Swiss catchments (Hari et al., 2006).

Given the past and future changes to Swiss river water temperatures and considering both the high sensitivity of aquatic species to river water temperatures and the increasing demand for river water by agriculture, industry and society as a whole, it is critical that we obtain a robust spatial and temporal understanding of the temperature increases that are expected for the many different rivers and streams of Switzerland. Here, we developed an efficient multi fidelity modelling method guided by statistical pattern recognition to estimate river water temperatures under climate change and thereby close the aforementioned spatial gap by determining, in an automated manner and on a country-wide scale, how future river water temperatures are likely going to change. By grouping catchments together via statistical pattern recognition, we were able to classify rivers (including spring-fed rivers) into 5 different thermal regimes, improving model results and enabling regime-specific analyses. The effect on warming by changing river discharge was investigate through a hysteresis analysis. Additionally, we introduce the *thermal extreme severity* index as an analytic tool to evaluate the change in thermal extreme amplitude.



## 2 Materials & Methods

A common challenge for model-based studies is the question of the optimal model to use. In surface hydrological applications, models can broadly be split into two major groups: process-based and statistical/stochastic models (Benyahya et al., 2007). Process-based models are based on physical equations and can resolve many hydrological processes in a physically robust manner, from the local to the catchment scale. However, albeit physically more robust, process-based models generally require a significant amount of input data and computational resources for the simulation of hydrological processes on the catchment scale, therefore limiting their applicability for climate change analyses on national scales. Statistical/stochastic models, as opposed to process-based models, are data driven, that is, are based on empirical relationships between input and output data. While they are physically less robust, their advantage lies in their relative simplicity and limited data requirements, sacrificing detail for increased repeatability and spatial cover. However, in order to build on the efficiency of statistics whilst preserving a clear physical basis, as a compromise between the two major groups, a sub-group of semi-empirical models, which employs physically meaningful equations but simplifies the more complex processes into purely empirical parameters, was developed (Piccolroaz et al., 2013). These semi-empirical models are ideally suited for hydrological climate change projections, as they provide much more robust projections compared to purely statistical approaches but simultaneously allow for a more comprehensive analysis than process-based models by enabling multi-model climate change ensemble analyses (La Fuente et al., 2022; Meehl et al., 2007).

In this study a novel multi-fidelity modelling approach able to choose from multiple different fidelity levels of two semi-empirical surface water temperature models, air2water and air2stream (Toffolon & Piccolroaz, 2015; Piccolroaz et al., 2013), was employed. Using multiple configurations on different levels of fidelity of two semi-empirical models allowed limiting the computational requirements to the levels needed for climate change ensemble simulations. The multi-fidelity approach, in which all available configurations (i.e., 3,4,5,6,7 and 8 different parameter combinations and implementations) of two different semi-empirical models were evaluated for their applicability to different thermal river regimes (Appendix A), allowed for developing optimal site-specific models for all the 82 thermal river monitoring stations of the Swiss Federal Office of the Environment (FOEN). As the driving model forcings (i.e., hydrological boundary conditions), we used downscaled near-surface air temperature projections from 22 coupled general circulation to regional climate models (GCM-RCM) from 9 GCM and 8 RCM, and combined them with projections of future stream discharge from 4 hydrological models for 3 climate change scenarios (i.e., representative concentration pathways) representing all climate protection measures with RCP2.6, moderate measures by RCP4.5, and business as usual by RCP8.5. Following recommendations from the Word Meteorological Organization (WMO, 2017) to use 30 years of continuous data while evaluating climate change, we selected 3 periods of interest including a reference period (1990 to 2019), a both near (2030 to 2059) and a far future period (2070 to 2099). Employing this multi-fidelity semi-empirical ensemble modelling approach enabled the production of nation-wide river temperature projections of unprecedented spatial coverage and uncertainty quantification. The method pathway is visualized in Figure 1.



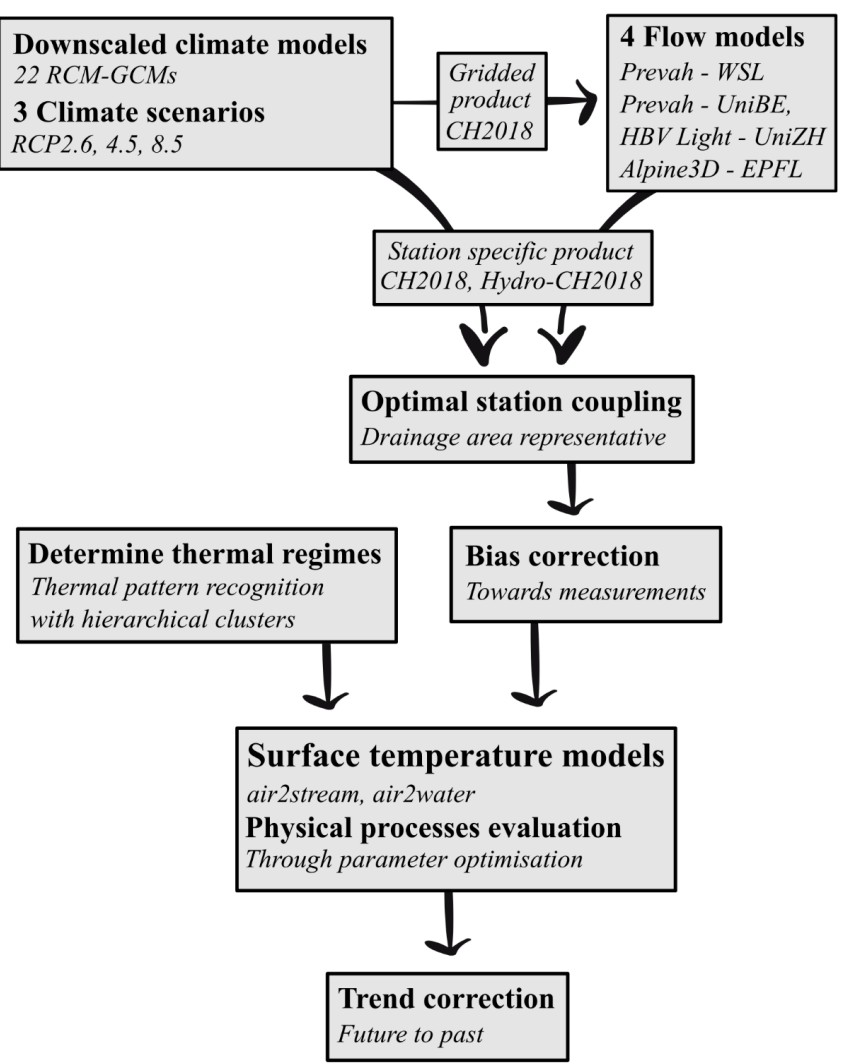

**Figure 1.** Workflow summarizing the data treatment and the multi-fidelity model selection and optimization.

## 2.1 Data

River water temperatures are directly influenced by both global and, to an even greater extent, local conditions in and above the drainage area, especially in regions divide by geographic barriers such as mountains (Ficklin et al., 2023). To analyze site-specific controls and project future river water temperatures, measured historic and simulated future climate data should thus be representative of the conditions and hydrologic processes upstream of the locations to be studied. The air2stream and air2water models require both measured historic and simulated future climate data to extend to at least a year (ideally more than one) and be daily resolved. However, to be sure that the effect of climate is included in calibration and analysis of future conditions, data should preferably cover 30 years (WMO, 2017; Piccolroaz et al., 2013).

Here, climate simulations for which near-surface air temperatures have been downscaled to local conditions with quantile mapping were used (CH2018, 2018). These data are available as both gridded and local station products (CH2018 Project Team, 2018). The gridded CH2018 version has been used to construct projections of future river discharge for 4 hydrological




models in the Hydro-CH2018 project (FOEN, 2021). The 4 models that were applied to
generate river discharge projections in the Hydro-CH2018 project are PREVAH-WSL ($M_1$;
Brunner, et al., 2019a; Brunner, et al., 2019b), PREVAH-UniBE ($M_2$; Muelchi et al., 2021),
HBV Light-UniZH ($M_3$; Freudiger et al., 2021), Alpine3D-EPFL ($M_4$; Michel et al., 2022)
(Figure 2a). The Hydro-CH2018 project produced projections for 61 out of the 82 FOEN river
monitoring stations under multiple different GCM-RCMs and 3 climate change scenarios
(RCP2.6, 4.5, and 8.5). The available projections, the employed circulation and hydrological
models, and the considered climate change scenarios for all the different stations that were
considered in this study are summarized in Table 1.

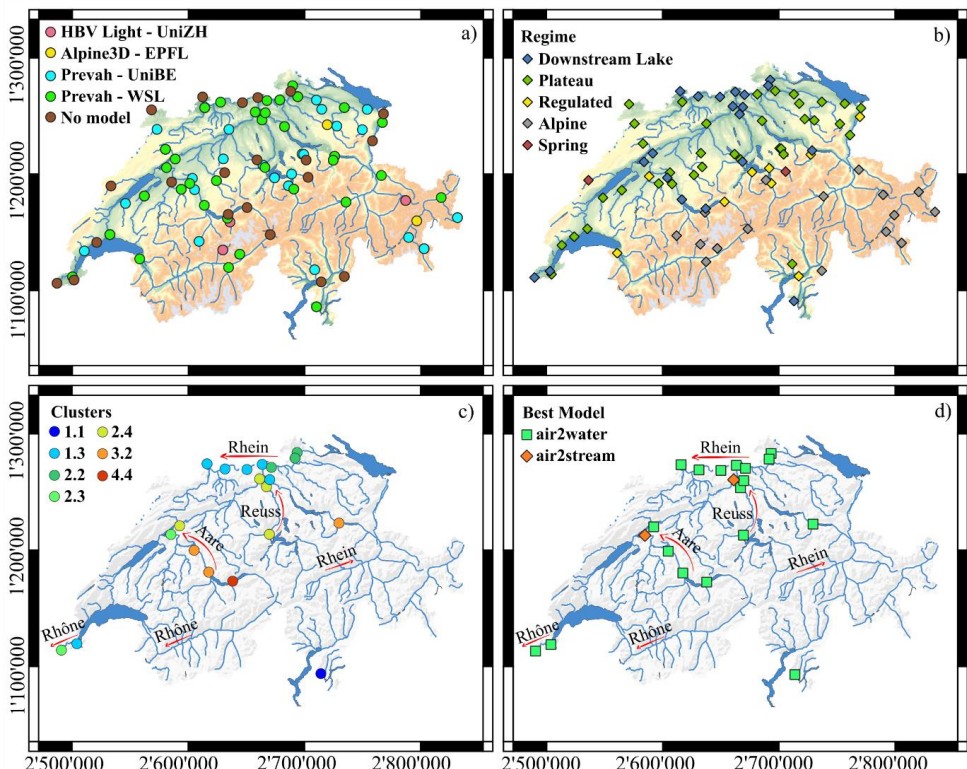

**Figure 2.** a) Investigated FOEN stations with available and used hydrological models providing future projections of river flow, b) station thermal regimes, c) downstream lake clusters, d) best performing surface water temperature model at downstream lake stations. Red arrows show river flow directions. Coordinate reference system is the Swiss LV95. Background map is the DHM25, swisstopo.admin.ch/de/geodata/height/dhm25.html).

From models $M_1$-$M_3$, continuous projections of river discharge at daily resolution for the entire
period covering 1990-2099 were available, projections from the $M_4$ model were discontinuous
and only covered the periods 1990-2000, 2005-2015, 2030-2040, 2055-2065, and 2080-2090.
River temperature simulations of river monitoring stations for which forcing data from models
$M_1$-$M_3$ were available covered the entire period of 1990-2099, while for stations for which only
data from model $M_4$ were available, simulations were only run for the periods for which data
was available.





**Table 1.** Climate projections and hydraulic models used for temperature simulation. For a complete climate model designation, see the CH2018 project report (CH2018, 2018). Models analyzed are indicated by an "X" mark, and models not analyzed but with simulation data provided by a "(X)" mark.

| GCM | RCM | PREVAH-WSL ($M_1$) | | | | | | PREVAH-UniBE ($M_2$) | | | | | |
|---|---|---|---|---|---|---|---|---|---|---|---|---|---|
| | | RCP8.5 | | RCP4.5 | | RCP2.6 | | RCP8.5 | | RCP4.5 | | RCP2.6 | |
| | | 0.11° | 0.44° | 0.11° | 0.44° | 0.11° | 0.44° | 0.11° | 0.44° | 0.11° | 0.44° | 0.11° | 0.44° |
| ICHEC-EC-EARTH | KNMI-RACMO22E | | X | | X | | | | X | | X | | |
| | DMI-HIRHAM5 | X | (X) | X | (X) | X | (X) | X | (X) | X | (X) | X | |
| | CLMcom-CCLM4-8-17 | | | | | | | X | | X | | | |
| | CLMcom-CCLM5-0-6 | | X | | | | | | X | | | | |
| | SMHI-RCA4 | X | (X) | X | (X) | X | (X) | X | (X) | X | (X) | X | (X) |
| MOHC-HadGEM2-ES | CLMcom-CCLM4-8-17 | | X | | | | | X | (X) | X | | | |
| | CLMcom-CCLM5-0-6 | | X | | | | | | X | | | | |
| | ICTP-RegCM4-3 | | | | | | | | | | | | |
| | KNMI-RACMO22E | | X | | X | | X | | X | | X | | X |
| | SMHI-RCA4 | X | (X) | X | (X) | X | X | X | (X) | X | (X) | X | X |
| MPI-M-MPI-ESM-LR | CLMcom-CCLM4-8-17 | | | | | | | X | (X) | X | (X) | | |
| | CLMcom-CCLM5-0-6 | | X | | | | | | X | | | | |
| | MPI-CSC-REMO2009-1 | | | | | | | X | (X) | X | (X) | X | (X) |
| | SMHI-RCA4 | X | (X) | X | (X) | | X | X | (X) | X | (X) | | X |
| | MPI-CSC-REMO2009-2 | | | | | | | X | (X) | X | (X) | X | (X) |
| MIROC-MIROC5 | CLMcom-CCLM5-0-6 | | X | | | | | | X | | | | |
| | SMHI-RCA4 | | X | | X | | X | | X | | X | | X |
| CCCma-CanESM2 | SMHI-RCA4 | | X | | X | | | | X | | X | | |
| CSIRO-QCCCE-CSIRO-Mk3-6-0 | SMHI-RCA4 | | | | | | | | X | | X | | |
| IPSL-IPSL-CM5A-MR | SMHI-RCA4 | | | | | | | X | (X) | X | (X) | | |
| NCC-NorESM1-M | SMHI-RCA4 | | X | | X | | X | | X | | X | | X |
| NOAA-GFDL-GFDL-ESM2M | SMHI-RCA4 | | | | | | | | X | | X | | |

| GCM | RCM | HBV Light-UniZH ($M_3$) | | | | | | Alpine 3D ($M_4$) | | | | | |
|---|---|---|---|---|---|---|---|---|---|---|---|---|---|
| | | RCP8.5 | | RCP4.5 | | RCP2.6 | | RCP8.5 | | RCP4.5 | | RCP2.6 | |
| | | 0.11° | 0.44° | 0.11° | 0.44° | 0.11° | 0.44° | 0.11° | 0.44° | 0.11° | 0.44° | 0.11° | 0.44° |
| ICHEC-EC-EARTH | KNMI-RACMO22E | | X | | X | | | | | | | | |
| | DMI-HIRHAM5 | X | | X | | X | | X | | X | | X | |
| | CLMcom-CCLM4-8-17 | X | | X | | | | | | | | | |
| | CLMcom-CCLM5-0-6 | | X | | | | | | | | | | |
| | SMHI-RCA4 | X | | X | | X | | X | | X | | X | |
| MOHC-HadGEM2-ES | CLMcom-CCLM4-8-17 | X | | X | | | | | | | | | |
| | CLMcom-CCLM5-0-6 | | X | | | | | | | | | | |
| | ICTP-RegCM4-3 | | X | | | | | | | | | | |
| | KNMI-RACMO22E | | X | | X | | X | | X | | X | | X |
| | SMHI-RCA4 | X | | X | | | X | | X | | X | | X |
| MPI-M-MPI-ESM-LR | CLMcom-CCLM4-8-17 | X | | X | | | | | | | | | |
| | CLMcom-CCLM5-0-6 | | X | | | | | | | | | | |
| | MPI-CSC-REMO2009-1 | | | | | | | | | | | | |
| | SMHI-RCA4 | X | | X | | | X | X | | X | | | X |
| | MPI-CSC-REMO2009-2 | X | | X | | X | | | | | | | |
| MIROC-MIROC5 | CLMcom-CCLM5-0-6 | | X | | | | | | | | | | |
| | SMHI-RCA4 | X | | X | | | X | | X | | X | | X |
| CCCma-CanESM2 | SMHI-RCA4 | X | | X | | | | | | | | | |
| CSIRO-QCCCE-CSIRO-Mk3-6-0 | SMHI-RCA4 | X | | X | | | | | | | | | |
| IPSL-IPSL-CM5A-MR | SMHI-RCA4 | X | | X | | | | | | | | | |
| NCC-NorESM1-M | SMHI-RCA4 | X | | X | | X | | | X | | X | | X |
| NOAA-GFDL-GFDL-ESM2M | SMHI-RCA4 | X | | X | | | | | | | | | |

| GCM | RCM | No Flow Projection | | | | | |
|---|---|---|---|---|---|---|---|
| | | RCP8.5 | | RCP4.5 | | RCP2.6 | |
| | | 0.11° | 0.44° | 0.11° | 0.44° | 0.11° | 0.44° |
| ICHEC-EC-EARTH | KNMI-RACMO22E | | X | | X | | |
| | DMI-HIRHAM5 | X | (X) | X | (X) | X | |
| | CLMcom-CCLM4-8-17 | X | | X | | | |
| | CLMcom-CCLM5-0-6 | | X | | | | |
| | SMHI-RCA4 | X | (X) | X | (X) | X | (X) |
| MOHC-HadGEM2-ES | CLMcom-CCLM4-8-17 | X | (X) | X | | | |
| | CLMcom-CCLM5-0-6 | | X | | | | |
| | ICTP-RegCM4-3 | | X | | | | |
| | KNMI-RACMO22E | | X | | X | | X |
| | SMHI-RCA4 | X | (X) | X | (X) | | X |
| MPI-M-MPI-ESM-LR | CLMcom-CCLM4-8-17 | X | (X) | X | (X) | | |
| | CLMcom-CCLM5-0-6 | | X | | | | |
| | MPI-CSC-REMO2009-1 | X | (X) | X | (X) | X | (X) |
| | SMHI-RCA4 | X | (X) | X | (X) | | X |
| | MPI-CSC-REMO2009-2 | X | (X) | X | (X) | X | (X) |
| MIROC-MIROC5 | CLMcom-CCLM5-0-6 | | X | | | | |
| | SMHI-RCA4 | | X | | X | | X |
| CCCma-CanESM2 | SMHI-RCA4 | X | | X | | | |
| CSIRO-QCCCE-CSIRO-Mk3-6-0 | SMHI-RCA4 | X | | X | | | |
| IPSL-IPSL-CM5A-MR | SMHI-RCA4 | X | (X) | X | | | |
| NCC-NorESM1-M | SMHI-RCA4 | X | | X | | X | |
| NOAA-GFDL-GFDL-ESM2M | SMHI-RCA4 | X | | X | | | |





Measurements of historic meteorologic and hydraulic parameters which were used for model
calibration, validation and for bias correction were obtained at daily resolution from the
MeteoSwiss IDAweb platform (meteoschweiz.admin.ch) and from the Hydrology Division of
the Federal Office for the Environment FOEN (hydrodaten.admin.ch). For monitoring stations
at which historic river discharge data or future river discharge projections weren't available,
only future near-surface air temperature projections were used to simulate water temperature.
Where climate projections were available at multiple different spatial resolutions (i.e. 0.11°
and 0.44°), only one model, as indicated in Table 1, was included in the analysis, following the
approach of Muelchi et al., 2021.

## 2.2 Hydrologic and meteorologic station coupling

Switzerland is characterized by a pronounced topography. Therefore, the closest
meteorological station to a hydraulic station might not necessarily be the ideal coupling partner.
Hydraulic and meteorological stations were instead paired according to the following
procedure: Only stations for which (a) future climate projections of near-surface air
temperatures (required) and river discharge (optional, but desirable for improved water
temperature predictions) were available for the entire period covering 1980 to 2099, and (b)
historic measurements of near-surface air temperatures and river discharge were available from
1980 to 2020, were considered. Meteorological stations were subsequently paired with
hydrological stations such that (a) the horizontal distance between river and meteorological
stations was minimal (criterion "DIS"), (b) the meteorological station was representative of the
conditions in the upstream drainage area (criterion "DRA"), and (c) the elevation difference
didn't exceed a reasonable threshold of 200 m (criterion "ELE"). Where possible, all three
criteria were adhered to. For situations where the closest meteorological station was either not
fulfilling DRA or ELE, the DIS criterion was evaluated only for stations which fulfilled both
DRA and ELE. Station details and pairings are summarized in Table 2.




**Table 2.** Combined river and meteorological stations and available models for climate projections of discharge. Abbreviations: DIS: Distance; ELE: Elevation; DRA: Drainage area.

| FOEN Hydrological stations | | | | Meteorological stations | | | | Hydrological models Hydro-CH2018 | | | |
|---|---|---|---|---|---|---|---|---|---|---|---|
| Name | ID | Height (m a.s.l.) | Area (km²) | Acrony | Height (m a.s.l.) | Distance (km) | Criteria | $M_1$ | $M_2$ | $M_3$ | $M_4$ |
| Rhône - Porte du Scex | 2009 | 377 | 5238 | AIG | 381 | 3.8 | DIS | X | | | |
| Aare - Brugg | 2016 | 332 | 1168 | BUS | 387 | 14.0 | DIS | X | | | |
| Reuss - Mellingen | 2018 | 345 | 3386 | BUS | 387 | 15.0 | DIS | X | | | |
| Aare - Brienzwiler | 2019 | 570 | 555 | MER | 588 | 6.1 | DIS | | | | |
| Aare - Brügg, Aegerten | 2029 | 428 | 8249 | BER | 553 | 20.0 | ELE | X | | | |
| Aare - Thun | 2030 | 548 | 2459 | INT | 577 | 22.3 | DIS | X | | | |
| Vorderrhein - Ilanz | 2033 | 693 | 774 | CHU | 556 | 26.9 | DRA | X | X | | |
| Broye - Payerne, Caserne d 'aviation | 2034 | 441 | 416 | PAY | 490 | 2.7 | DIS | X | X | | X |
| Thur - Andelfingen | 2044 | 356 | 1702 | SHA | 438 | 11.4 | DIS | X | X | X | |
| Reuss - Seedorf | 2056 | 438 | 833 | ALT | 438 | 0.4 | DIS | X | X | | |
| Ticino - Riazzino | 2068 | 200 | 1613 | MAG | 203 | 1.8 | DIS | | | | |
| Emme - Emmenmatt, nur Hauptstation | 2070 | 638 | 443 | LAG | 744 | 4.7 | DIS | X | X | | |
| Muota - Ingenbohl | 2084 | 438 | 317 | ALT | 438 | 12.8 | DIS | | X | | |
| Aare - Hagneck | 2085 | 437 | 5112 | BER | 553 | 22.5 | DRA | X | | | |
| Rhein - Rheinfelden, Messstation | 2091 | 262 | 3452 | BAS | 316 | 16.4 | DIS | X | | | |
| Linth - Weesen, Biäsche | 2104 | 419 | 1062 | GLA | 517 | 10.9 | DIS | X | X | | |
| Birs - Münchenstein, Hofmatt | 2106 | 268 | 887 | BAS | 316 | 3.7 | DIS | X | X | | X |
| Lütschine - Gsteig | 2109 | 585 | 381 | INT | 577 | 0.9 | DIS | X | | X | X |
| Sitter - Appenzell | 2112 | 769 | 74.4 | STG | 776 | 10.4 | DIS | | X | | |
| Aare - Felsenau, K.W. Klingnau (U.W.) | 2113 | 312 | 1768 | BUS | 386 | 25.8 | DRA | | | | |
| Murg - Wängi | 2126 | 466 | 80.2 | TAE | 539 | 4.1 | DIS | | X | | |
| Rhein (Oberwasser) - Laufenburg | 2130 | 299 | 3405 | RUE | 611 | 18.6 | DIS | | | | |
| Aare - Bern, Schönau | 2135 | 502 | 2941 | BER | 553 | 6.5 | DIS | X | | | |
| Rheintaler Binnenkanal - St. Margrethen | 2139 | 404 | 175 | VAD | 457 | 37.3 | DRA | | | | |
| Rhein - Rekingen | 2143 | 323 | 1476 | KLO | 426 | 18.5 | DRA | X | | | |
| Landquart - Felsenbach | 2150 | 571 | 614 | RAG | 497 | 9.5 | DIS | X | | | |
| Reuss - Luzern, Geissmattbrücke | 2152 | 432 | 2254 | LUZ | 454 | 2.0 | DIS | X | | | |
| Gürbe - Belp, Mülimatt | 2159 | 522 | 116.0 | BER | 553 | 12.1 | DIS | | X | | |
| Massa - Blatten bei Naters | 2161 | 1446 | 196 | GRC | 1605 | 24.9 | ELE | X | | X | |
| Tresa - Ponte Tresa, Rocchetta | 2167 | 268 | 609 | LUG | 273 | 9.1 | DIS | X | X | | |
| Arve - Genève, Bout du Monde | 2170 | 380 | 1973 | GVE | 410 | 7.9 | DIS | | | | |
| Rhône - Chancy, Aux Ripes | 2174 | 336 | 1030 | GVE | 411 | 16.0 | DIS | | | | |
| Sihl - Zürich, Sihlhölzli | 2176 | 412 | 343 | SMA | 556 | 3.2 | DIS | X | X | | |
| Sense - Thörishaus, Sensematt | 2179 | 553 | 351 | BER | 553 | 14.3 | DIS | X | X | | |
| Thur - Halden | 2181 | 456 | 1085 | GUT | 440 | 11.8 | DIS | X | X | | |
| Doubs - Ocourt | 2210 | 417 | 1275 | FAH | 596 | 13.0 | DIS | | X | | |
| Allenbach - Adelboden | 2232 | 1297 | 28.8 | ABO | 1321 | 0.9 | DIS | | X | | |
| Limmat - Baden, Limmatpromenade | 2243 | 351 | 2384 | REH | 444 | 16.6 | DIS | X | | | |
| Rosegbach - Pontresina | 2256 | 1766 | 66.5 | SAM | 1709 | 4.3 | DIS | | X | | |
| Inn - Tarasp | 2265 | 1183 | 1581 | SCU | 1304 | 0.6 | DIS | X | | | |
| Lonza - Blatten | 2269 | 1520 | 77.4 | GRC | 1605 | 24.9 | ELE | | | X | X |
| Grosstalbach - Isenthal | 2276 | 767 | 43.9 | ALT | 438 | 5.3 | DIS | | | X | X |
| Sperbelgraben - Wasen, Kurzeneialp | 2282 | 911 | 0.56 | NAP | 1403 | 7.5 | DIS | | | | |
| Rhein - Neuhausen, Flurlingerbrücke | 2288 | 383 | 1193 | SHA | 438 | 0.9 | DIS | X | | | |
| Areuse - St-Sulpice | 2290 | 755 | 104 | BRL | 1050 | 9.0 | DRA | | | | |
| Suze - Sonceboz | 2307 | 642 | 127 | CHA | 1594 | 11.5 | DIS | X | X | | X |
| Goldach - Goldach, Bleiche, nur Hauptstation | 2308 | 399 | 50.4 | GUT | 440 | 19.3 | ELE | | X | | |
| Dischmabach - Davos, Kriegsmatte | 2327 | 1668 | 42.9 | DAV | 1594 | 4.9 | DIS | | | X | X |
| Langeten - Huttwil, Häberenbad | 2343 | 597 | 59.9 | WYN | 422 | 15.0 | DIS | | X | | |
| Riale di Roggiasca - Roveredo, Bacino di | 2347 | 980 | 8.12 | GRO | 323 | 6.0 | DIS | | | | |
| Vispa - Visp | 2351 | 659 | 786 | VIS | 639 | 3.6 | DIS | X | | | |
| Poschiavino - La Rösa | 2366 | 1860 | 14.1 | BEH | 2260 | 3.8 | DIS | | X | X | |
| Mentue - Yvonand, La Mauguettaz | 2369 | 449 | 105.0 | PAY | 490 | 17.1 | ELE | | X | | |
| Linth - Mollis, Linthbrücke | 2372 | 436 | 600 | GLA | 517 | 7.4 | DIS | X | X | | |
| Necker - Mogelsberg, Aachsäge | 2374 | 606 | 88.1 | EBK | 623 | 10.1 | DIS | | X | | |
| Murg - Frauenfeld | 2386 | 390 | 213 | TAE | 539 | 9.9 | DIS | | X | | |
| Rhein (Oberwasser) - Rheinau | 2392 | 353 | 1195 | SHA | 438 | 5.8 | DIS | | | | |
| Liechtensteiner Binnenkanal - Ruggell | 2410 | 435 | 116 | VAD | 457 | 12.9 | DIS | | | | X |
| Rietholzbach - Mosnang, Rietholz | 2414 | 682 | 3.19 | EBK | 623 | 13.5 | DIS | | | | X |
| Glatt - Rheinsfelden | 2415 | 336 | 417 | KLO | 426 | 11.4 | DIS | X | X | | |
| Venoge - Ecublens, Les Bois | 2432 | 383 | 228.0 | PUY | 456 | 9.2 | DIS | X | X | | |
| Aubonne - Allaman, Le Coulet | 2433 | 390 | 105 | CGI | 458 | 15.9 | DIS | | | | |
| Dünnern - Olten, Hammermühle | 2434 | 400 | 234 | WYN | 422 | 13.3 | DRA | | X | | |
| Aare - Ringgenberg, Goldswil | 2457 | 564 | 1138 | INT | 577 | 2.5 | DIS | | | | |
| Inn - S-Chanf | 2462 | 1645 | 616 | SAM | 1708 | 13.3 | DIS | | | | X |
| Saane - Gümmenen | 2467 | 473 | 1881 | BER | 552 | 17.6 | DIS | | | | |
| Rhein - Diepoldsau, Rietbrücke | 2473 | 410 | 6299 | VAD | 457 | 29.9 | DRA | X | | | |
| Engelberger Aa - Buochs, Flugplatz | 2481 | 443 | 228 | LUZ | 454 | 10.6 | DIS | | X | X | |
| Allaine - Boncourt, Frontière | 2485 | 366 | 212 | FAH | 596 | 10.1 | DIS | | | | |
| Promenthouse - Gland, Route Suisse | 2493 | 394 | 120 | CGI | 458 | 3.2 | DIS | | X | | |
| Schlichenden Brünnen - Muotathal | 2499 | 638 | 31 | ALT | 437 | 15.6 | DIS | | | | |
| Worble - Ittigen | 2500 | 522 | 67.1 | BER | 553 | 2.2 | DIS | | X | | |
| Biber - Biberbrugg | 2604 | 825 | 31.9 | EIN | 911 | 3.5 | DIS | | X | | |
| Rhône - Genève, Halle de l 'île | 2606 | 367 | 8000 | GVE | 411 | 4.9 | DIS | X | | | |
| Sellenbodenbach - Neuenkirch | 2608 | 515 | 10.4 | LUZ | 454 | 11.0 | DIS | | | | |
| Alp - Einsiedeln | 2609 | 840 | 46.7 | EIN | 911 | 2.4 | DIS | | X | | |
| Riale di Pincascia - Lavertezzo | 2612 | 536 | 44.5 | OTL | 367 | 10.4 | ELE | | X | | |
| Rhein - Weil, Palmrainbrücke | 2613 | 244 | 3645 | BAS | 316 | 6.7 | DIS | | | | |
| Rom - Müstair | 2617 | 1236 | 128 | SMM | 1386 | 0.4 | DIS | | X | X | |
| Rhône - Oberwald | 2623 | 1368 | 93.3 | ULR | 1345 | 4.6 | DRA | | | | |
| Kleine Emme - Emmen | 2634 | 430 | 478 | LUZ | 454 | 4.2 | DIS | | X | X | X |
| Grossbach - Einsiedeln, Gross | 2635 | 942 | 8.95 | EIN | 910 | 3.0 | DIS | | | | |





### 2.3 Forcing data bias correction

Differences between near-surface air temperature measurements used for calibration and climate model projections, even when slight, may artificially alter the quantification of projected future river water temperatures by introducing a systematic bias at the start of the simulations. Despite the fact that the highly resolved GCM-RCMs model output data products that were considered here were already statistically downscaled, small differences between modelled and observed air temperatures during the reference period could still be detected. For the river discharge projections, no bias correction has so far been performed. To mitigate this bias, the time series of air temperatures and river discharge used as climate forcing data were statistically adjusted using the change factor method (Diaz-Nieto & Wilby, 2005; Minville et al., 2008). This method adjusts climate projections towards measurements by removing the climatological year (consisting of daily averages) from first the modeled data and then adding the corresponding climatological year from measurements according to Eq. 1, thereby correcting long-term and seasonal biases while maintaining individual climate model trends and stochastic variabilities.

$$Fn_i = \left(Fo_i - Co_j\right) + Cm_j \tag{1}$$

where $Fn_i$ is the adjusted variable at time $i$, $Fo_i$ is the future climate simulated time series of either air temperatures or river discharge at daily resolution, and $Co_j$ and $Cm_j$ are the climatological years of the climate simulated time-series and the historic measurements, respectively, at the day of year $j$ corresponding to time $i$. The climatological years were smoothed using a 60-day window to remove the effect of possible pulse events, especially for discharge. Due to low flow conditions in some rivers, discharge in these rivers was never adjusted below the minimum observed flow.

### 2.4 Thermal regime classification

For the multi-fidelity modelling approach, the different river monitoring stations were re-classified into the 4 different thermal regimes that have previously been identified for Switzerland (Michel et al., 2020; Piccolroaz et al., 2016) as well as 1 additional thermal regime defined for the purpose of this study. The existing thermal regimes are "Downstream Lake", "Swiss Plateau", "Alpine", "Regulated", while the "Spring" discharge regime was added to address the special thermal case of stations situated at the mouth of spring fed streams. "Downstream Lake" stations show a clear de-coupling between river temperature and river discharge, "Swiss Plateau" stations exhibit an annual flow cycle with minimal discharge in summer and strong interannual variability, "Alpine" stations show that both discharge and temperature are strongly influenced by snow and glacier melt, "Regulated" stations are fed by intermittent releases of large volumes of water from upstream reservoirs, and "Spring" stations located immediately downstream of springs and characterized by a nearly constant temperature signal decoupled from air temperature.

The already existing classifications from (Michel et al., 2020; Piccolroaz et al., 2016) and the suitability of the yet unclassified stations to be grouped under the different thermal regimes were first explored by evaluating the historic data and the location visually (Figure 2b). Following this first visual classification, an automated thermal pattern recognition using hierarchical clusters via the multi-cluster tool DTWARP_PER_33 (Bögli, 2020) was used (Figure 2c). Application of the thermal pattern recognition matched the visual pre-classification in most instances, but revealed that, for certain stations located far downstream of lakes, upstream lake processes are still the dominant control for river water temperatures. Stations that were previously classified as not being part of the Downstream Lake regime were thus



here reclassified as Downstream Lake according to the results of the thermal pattern recognition procedure.

At Downstream Lake stations, multiple configurations of both water temperature models (air2stream and air2water) were tested through calibration, and only the best performing temperature model and parameter setup was kept (station thermal regimes as well as cluster results are shown in Figure 2 and provided in Table B1). For the remaining stations not belonging to the Downstream Lake regime, river processes such as local flow variations and water depth dominate the water temperature development. For these stations, different model configurations of only the air2stream model were explored.

## 2.5 Surface water temperature model setup

Two semi-empirical surface water temperature models were employed, the river water model air2stream (Toffolon & Piccolroaz, 2015)[*1] and the lake water model air2water (Piccolroaz et al., 2013)[*2], with the former being an extended version of the latter. air2stream and air2water combine the simplicity of stochastic models with accurate empirical representation of the relevant physical processes affecting water temperature. Both models require near-surface air temperature as input to predict future river temperature, while discharge may be incorporated in air2stream to further improve river temperature predictions but isn't required.

Both models include up to eight parameters ($a_1$ to $a_8$) which are fitted towards measured data. Apart from the effect of air temperature on water temperature, the models additionally resolve the effect of river depth, discharge, thermal different tributaries, invers stratification in lakes during winter, and seasonal cycles. Model complexity, i.e. how many processes are directly being resolved by the models or indirectly included through parameter estimation, can be varied by removal of one or more of the additional processes listed above, resulting in the use of 8, 7, 6, 5, 4 or 3 parameters. Depending on local conditions, model performance can be improved by the removal of processes which plays a minor or insignificant role for water temperature, thereby the need to correctly chose model complexity. For additional information about air2stream and air2water see Appendix A and Piccolroaz et al. (2013) and Toffolon & Piccolroaz (2015).

For the simulation of future river temperatures, a multi-fidelity modelling approach that identified the best water temperature model for each single river monitoring station that was considered in this study was employed. The optimal model parameter configuration for each station was identified via a Monte-Carlo calibration process performed with the Crank Nicolson scheme (Crank & Nicolson, 1947), consisting of over 2'000 runs using Particle Swarm Optimization (Kennedy & Eberhart, 1995) with 500 particles. The Root Mean Square Error (RMSE) function was used as the objective function and combined with the *dotty-plots* quality check (S. Piccolroaz et al., 2013; Piccolroaz, 2016; Toffolon et al., 2014).

Temporally overlapping, daily averaged near-surface air temperature and river discharge measurements spanning the 30-year reference period of 1990 to 2020 were used as calibration data, while for validation the data from 1980 to 1990 were used. By choosing to use the most recent data for calibration rather than validation ensures that recent local climate conditions are carried into future projections (Shen et al., 2022). For the few cases where no forcing data for calibration did exist between 1990 to 2020 (Table C2), validation was deprioritized and calibration done on the 1980-1990 data. For stations missing either historical data or future

---

[*1] github.com/marcotoffolon/air2stream
[*2] github.com/marcotoffolon/air2water





projections of river discharge (brown markers, Figure 2a), discharge was not considered as
forcing data and the air2stream model was reduced to a 3 or 5 parameter model, while no
adaptation was required for air2water as it doesn't simulate discharge. Datasets used for
calibration and validation with data gaps shorter than 30 days were filled via linear
interpolation, while for datasets with gaps exceeding 30 days only the longest continuous
dataset was used.

All simulations (calibration, validation and climate runs)  used a one year period as a spin-up
with the first year of forcing data repeated. Only the best performing river temperature model
was considered for the follow on climate runs. The final calibration and validation periods and
the best performing parameter setups for each station are provided in Table B2. As initial
conditions for the stepwise climate simulations with model $M_4$, we used simulated temperature
from the latest prior simulated date, that is, climate simulations between 2030 to 2040 used
temperature from end of 2015 as initial condition.

### 2.6 Trend correction

Empirical models generally predict less warming in the future compared to physically based
models, the primary reason being underrepresentation of the thermal catchment memory,
including snow and ice (Leach & Moore, 2019). To quantify how good the models air2stream
and air2water, which both lack deterministic considerations of snow and ice melt, are able to
recreate past trends, we compared trends from river water temperature measurements and
corresponding modeled temperature trends between 1990 and 2019. On an annual basis, this
comparison was possible for 25 out of 82 stations, consisting of 9 Downstream Lake, 7
Regulated, 7 Swiss Plateau, 2 Alpine, and 0 Spring regime stations. Stations were selected with
a 30 years of continuous data requirement in air and water temperature and river discharge.
Only statistically significant trends ($p < 0.05$) were considered.

Both air2stream and air2water underestimate the annual temperature trend during the reference
period on average by 0.14 and 0.11 °C per decade, respectively. For air2stream, the annual
trend bias is smallest for the Swiss Plateau regime (0.09 °C per decade) and largest in the
Alpine regime (0.17 °C per decade). Seasonally, the trend bias is largest from June to August
and September to November, whereas, especially for air2water, the bias is small from
December to February and March to May.

The divergence of both air2stream and air2water from observed trends warrant a post
simulation bias correction of simulated trends. The bias is station dependent, making an
individual correction at each station preferable (Tables B3 to B6). However, only about 30%
of the stations investigated have long enough data sets (30 years) for individual correction.
Therefore, we tied the seasonal trend bias correction to the thermal regime, thereby keeping
the correction linked to local conditions. Note that no station of the Spring thermal regime had
enough data to allow for the trend bias correction. Spring stations were therefore not trend bias
corrected. As the trend bias correction is acting on climate simulations of river temperature
stretching from 1990 to 2099, the bias correction had to be scaled towards how air temperature
trends shift in the climate models. The scaling was designed such that it didn't affect the bias
correction during the reference period (1990 to 2019), while adjusting the correction towards
how the air temperature trend (*TTair*) changes in the near (2030 to 2059) and far future (2070
to 2099). For this purpose an adjustment factor *Fs* (-) was constructed from the mean climate
models air temperature trends for each climate scenario. *Fs* is thus specific for each climate
scenario, station and season.





$$Fs_{i,s} = \frac{TTair_{i,s}}{TTair_{ref,s}} \qquad (2)$$

Here $TTair_{i,s}$ is the mean of the air temperature trends from the climate models, which is changing for each season and with the reference, near, and far future periods, $TTair_{ref,s}$ is the mean of the seasonal air temperature trend during the reference period, $i$ is the number of days, and $s$ denotes the season. The temporal gaps between 1990 to 2019 to 2030 to 2059 and 2070 to 2099, during which the air temperature trends were calculated, were linearly filled with shape-preserving piecewise cubic interpolation resulting in a continuous $Fs_{i,s}$ from 1990 to 2099. $Fs_{i,s}$ varied from -2 to +3 depending on the season and climate scenario and was applied for simulations using discharge input from models $M_1$ to $M_3$, while for simulations using $M_4$, $Fs_{i,s}$ was set to 1 from 1990 to 2099 due to too short simulation time frames in $M_4$ (only one decade). With $Fs_{i,s}$, the seasonal and thermal regime dependent water temperature bias $Tb_{i,s}$ (regime dependent mean from Table C3 to C6) is turned into the thermal regime and climate scenario dependent seasonal bias correction $Bc_s$ (°C day$^{-1}$)

$$Bc_s = \sum_{i=1}^{i=n} Fs_{i,s} * Tb_{i,s} \qquad (3)$$

where $n$ is the number of days since 1$^{st}$ of January 1990. Before adjusting the water temperature model output from 1990 to 2099, $Bc_s$ was combined into a continuous dataset by filling in the 3- to 5-day gap in between each season with shape-preserving interpolation. The trend adjustment applied here with $Fs$, $Bc$, and pre- and post-adjustment data is shown from one example station in Figure B1. Pre and post trend correction for the difference in modeled and measured trends is summarized in Table B7.

## 2.7 Thermal hysteresis

Hysteresis, wherein a dependent variable (water temperature or suspended sediments) can exhibit multiple values in response to a single value from the independent variable (discharge), is a common phenomenon in hydrology (Gharari & Razavi, 2018). Hysteresis can be caused in rivers by emptying and refiling of sediment layers (Tananaev, 2012), or as a lag in stream temperature response to air temperature caused by ice-melt or reservoir release (Van Vliet et al., 2011; Webb & Nobilis, 1994).

We investigated past and future hysteresis loops between water temperatures (the dependent variable) and river discharge (the independent variable) using a versatile index (Zuecco index, Zuecco et al., 2016). The index divides loops into 8 classes (I to VIII) depending on rotation direction (counter clockwise or clockwise), number of loops and loop sizes. The Zuecco index works through the computation of definite integrals on data in chosen intervals and was developed for hysteretic loops where the independent variable increases from its initial value, reaches a peak and then decreases.




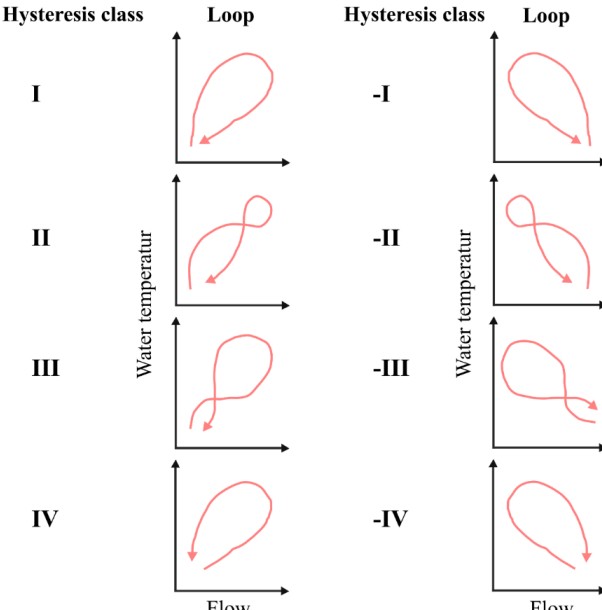

**Figure 3**. Hysteresis classes with corresponding hysteresis loops. Expanded with classes -I to -IV from Zuecco et al., (2016) to incorporate water temperature as the dependent variable.

Here, only classes I to IV is fitted to the data. Moreover, in lowland rivers in Switzerland,
discharge in winter can be larger than in spring or summer, an effect enhanced by ongoing
climate warming through shortening or elimination of snow cover and glacial melt (FOEN
(ed.), 2021; Michel et al., 2020; Van Vliet et al., 2013). To incorporate this reversed hysteretic
loop, we added 4 "mirrored" hysteresis classes, -I to -IV, to the 8 introduced by Zuecco et al.,
(2016) (Figure 3). This was done by inverting the normalized flow prior to the computation of
definite integrals, thus creating an increasing and decreasing independent variable. Post
inversion, the index thus gives class I to IV, but since the independent variable had been
inverted, it is shown here as -I to -IV. Note that the index works on set intervals. If the loops
do not come back to their initial values, it works with open loops. The length of the data sets
being investigated should depend on the quality and resolution of the data and the rate at which
the dependent variable changes with respect to the independent variable (Zuecco et al., 2016).
Here we used daily resolved datasets, averaged from 30 years of modeled data, thus always
providing full annual loops.

2.8 Temperature extremes
Extreme conditions are not straight forward to define. In general, they depend on what is
considered to be extreme in relation to normal conditions (Stephenson, 2008). A widely used
concept defines events as extreme if they are below or above the 10[th] or 90[th] percentile in a
distribution (IPCC, 2014). Here, water temperatures are considered to be extremely high if they
exceed the 90[th] percentile during the 30-year reference, near- and far-future periods.

We define a new "extreme event severity index", as the temperature difference between the
90[th] percentile to the median for each climate simulation and period. If this temperature gap
increases, it indicates that extreme temperatures become more severe as thermal peaks are
elevated compared to the median temperature. The severity of thermal extremes for each
simulation and period is thus X °C from 0 °C, where X denotes the difference between the 90[th]





percentile and the median temperature while 0 °C represent a match to the median temperature.
Our analysis was made independent of where (beginning or end) in the 30-year periods it was
conducted by removing the climatic trend for each simulation and period before calculating the
index. Note that by defining extreme events with the 90$^{th}$ percentile during each analyzed
period, we take into account temporal in-situ extreme events as they are experienced during
the considered periods. We do not inflate our results by using past extreme event definitions to
evaluate future extreme events.

## 2.9 Thermal Thresholds

By counting the number of days per year during which thermal thresholds are exceeded, effects
of climate change on fish can be evaluated both locally and regionally (Michel et al., 2020).
The occurrence of exceedance of specific river water temperature thresholds on a daily scale
was used to investigate the historic past (1990 to 2019) and projected future (2070 to 2099)
stress on the brown trout (*Salmo trutta*). Three thermal thresholds were chosen in order to
incorporate important aspects in the life of the brown trout. including: (1) adult mortality as
represented by a daily mean temperature above 25 °C (Elliott, 1981; Wehrly et al., 2007), also
set as a hard upper limit for the thermal use of waters in Switzerland (Water Protection
Ordinance 814.201); (2) an increased risk for proliferative kidney disease (PKD) as parasite
activity as represented by a daily mean temperature above 15 °C (Chilmonczyk et al., 2002;
Strepparava et al., 2018) and; (3) fish egg (roe) mortality from September to January as
represented by a daily mean temperature above 13 °C (Elliott, 1981).

# 3 Results

## 3.1 Warming

The most influential factor for future river water temperatures was the climate change
scenarios. Individual station warming, from the reference (1990-2019) to the near (2030-2059)
and far future (2070-2099) periods, is shown in Figure 4. Under the RCP8.5 scenario, the
warming of river temperatures increases throughout the 21$^{st}$ century, and even accelerates. The
smallest change in river temperatures was observed under the RCP2.6 scenario, with warming
reaching a plateau in the middle of the 21$^{st}$ century. The mean change in river temperatures
from the reference period to the near and far future amounts to +0.77 and +0.91 °C for RCP2.6,
to +0.95 and +1.51 °C for RCP4.5, and to +1.22 and +3.18 °C for RCP8.5, respectively. This
amounts to an averaged water warming rate from 1990 to 2099 for RCP8.5 of 0.36 °C per
decade, 0.19 °C per decade for RCP4.5, and 0.12 °C per decade for RCP2.6. At the same time
as near-surface air temperature changed by 0.50 °C per decade for RCP8.5, 0.26 °C per decade
for RCP4.5 and 0.13 °C per decade for RCP2.6.

Climate change impact was heterogeneous between stations, yet common patterns were found
within thermal regimes (Figure 4, Table B8). The strongest river water warming, regardless of
climate scenario or time period, was observed for stations in the Alpine regime, followed in
order by Downstream Lake, Regulated, Swiss Plateau, and Spring stations. Under RCP8.5,
river temperatures of Alpine stations, on average, warm by 1.44 °C until the near and by 3.54
406  °C until the far future, compared to the reference period. The river water of Downstream Lake
stations also strongly warmed, by 1.36 °C until the near and by 3.43 °C until the far future.
Compared to the Alpine and Downstream Lake thermal regimes, river temperatures of stations
in the Regulated (near future +1.19 °C, far future +3.00 °C) and Swiss Plateau (near future
+1.06 °C, far future +2.75 °C) regimes warmed less. Least affected, by a wide margin, were
the river temperatures of the 2 stations that classify as the Spring thermal regime (near future
+0.04 °C, far future +0.10 °C).



**Figure 4.** Modeled mean river temperature increase from the reference (1990 to 2019), to near (2030 to 2059, blue bars) and far future (2070 to 2099, red bars) under climate scenarios RCP 2.6, RCP4.5 and RCP8.5. Shown is the median (bar center line) and the lower and upper quartiles (left and right bar extent) of the difference between periodic mean temperatures (over 30 years) for each available climate simulation (additionally averaged where multiple hydrological models exist), i.e., the bar extents show climate model variability in the mean temperature change between the three periods. Stations 2414 and 2462 are not shown since the flow model $M_4$ lacked 30 years of continuous data.





## 3.2 Hysteresis analysis

The hysteresis class could be determined at for each station for with future and present river discharge (47 out of 82 stations). For all stations, climate scenarios, and climate models, the index found solutions in hysteresis intervals ranging from 328 to 164 days.

During the reference period the dominant class was IV (45.6%) followed by III (25.0%), -I (14.7%), -II (11.8%) and I (2.9%) while no stations belonged to class II. For the reference period the classes remained independent of climate scenario (RCP8.5, 4.5, 2.6) or hydrological model (M1, M2,M3) used, while in the near and far future differences start to show. For RCP8.5 in the far future period the dominant class was -I (48.5%) followed by class IV (33.8%), III (13.2%) and -II (4.4%).

For the RCP8.5 scenario classes is shown for the reference, near and far future periods in Table 3 (hysteresis classes for RCP4.5 are shown in Table B9, and for RCP2.6 in Table B10). Under RCP8.5, the number of stations which changed hysteresis classes between the reference and the near future was 23%, increasing to 51% until the far future. Correspondingly, under RCP4.5 23% had changed classes when reaching the near future, while 38% of the stations changed classes until the far future. Under RCP2.6, 28% of stations had changed classes until the near future, but once reaching the far future, some stations changed back again and the fraction of stations that were in a different hysteresis class compared to the reference period was reduced to 21%.

Considering only the far future, stations belonging to the Swiss Plateau thermal regime showed the largest change in hysteresis loop classes, with 58% changing under RCP8.5, 42% under RCP4.5 and 12% under RCP2.6. Considering again only the far future, stations belonging to the Regulated thermal regime exhibited hysteresis loop class changes of 50% under RCP8.5, 33% under RCP4.5 and 50% under RCP2.6. Least prone to hysteresis class changes in the far future were stations of the Alpine thermal regime (38% under RCP8.5 and RCP4.5, 23% under RCP2.6). Out of the 20 Downstream Lake thermal regime stations only 2 stations were investigated with discharge (i.e. model with air2stream instead of air2water). From these 2 stations, 1 changed hysteresis class with RCP8.5 by the far future, 1 with RCP2.6 but none with RCP4.5. As can be seen from 4 representative stations for the Swiss Plateau, Regulated, Alpine, and Downstream Lake illustrated in Figure 5, a change in hysteresis class is usually associated with a counterclockwise rotation and stretching of the loop from example a lower class to a higher class (III to IV). Such a rotation and stretching appears as a result of increased warming in summer combined with a decrease in summer discharge, while warming in winter is smaller than in summer and discharge is increasing.





**Table 3.** Change in hysteresis classes marked by yellow from the reference period (1990 to 2019) to the near (2030 to 2059) and the far future (2070 to 2099) for climate scenario RCP8.5. Flow data from models $M_2$, $M_3$ and $M_4$. Stations with no flow measurements for calibration, missing flow model output as forcing or where the use of the air2water model did not require flow as input have been excluded. A change in class from the reference period to the near or far future period is highlighted in *italic*.

| Station | Reference | | | Near | | | Far | | |
|---|---|---|---|---|---|---|---|---|---|
| | $M_1$ | $M_2$ | $M_3$ | $M_1$ | $M_2$ | $M_3$ | $M_1$ | $M_2$ | $M_3$ |
| **Downstream Lake** | | | | | | | | | |
| 2016 | 4 | | | 4 | | | *-1* | | |
| 2085 | 4 | | | 4 | | | 4 | | |
| **Regulated** | | | | | | | | | |
| 2009 | 3 | | | *4* | | | *4* | | |
| 2056 | 3 | 3 | | *4* | *4* | | *4* | *4* | |
| 2084 | | 4 | | | 4 | | | 4 | |
| 2372 | 4 | 4 | | 4 | 4 | | 4 | 4 | |
| 2473 | 3 | | | *4* | | | *4* | | |
| 2481 | | 4 | 4 | | 4 | 4 | | 4 | 4 |
| **Swiss Plateau** | | | | | | | | | |
| 2034 | -2 | -2 | | -2 | -2 | | -2 | *-1* | |
| 2044 | 4 | 4 | 4 | *-2* | *-1* | *-2* | *-1* | *-1* | *-1* |
| 2070 | 4 | 4 | | 4 | 4 | | *-1* | *-1* | |
| 2106 | -2 | -2 | | -2 | -2 | | -2 | *-1* | |
| 2112 | | 4 | | | 4 | | | 4 | |
| 2126 | | -1 | | | -1 | | | -1 | |
| 2159 | | 4 | | | 4 | | | *-1* | |
| 2176 | 4 | 4 | | 4 | 4 | | *-1* | *-1* | |
| 2179 | 4 | 4 | | 4 | 4 | | *-1* | *-1* | |
| 2181 | 4 | 4 | | 4 | 4 | | *-1* | *-1* | |
| 2210 | | -2 | | | -2 | | | | |
| 2307 | -1 | -1 | | -1 | -1 | | -1 | -1 | |
| 2308 | | 4 | | | *-1* | | | *-1* | |
| 2343 | | -1 | | | -1 | | | -1 | |
| 2369 | | -1 | | | -1 | | | -1 | |
| 2374 | | 4 | | | *-1* | | | *-1* | |
| 2386 | | -2 | | | *-1* | | | *-1* | |
| 2415 | -2 | -2 | | -2 | -2 | | -2 | *-1* | |
| 2432 | -1 | -1 | | -1 | -1 | | -1 | -1 | |
| 2434 | | -1 | | | -1 | | | -1 | |
| 2493 | | -1 | | | -1 | | | -1 | |
| 2500 | | -1 | | | -1 | | | -1 | |
| 2604 | | 4 | | | 4 | | | *-1* | |
| 2609 | | 4 | | | 4 | | | 4 | |
| 2612 | | 3 | | | 3 | | | 3 | |
| 2634 | | 4 | 4 | | 4 | 4 | | *-1* | *-1* |
| **Alpine** | | | | | | | | | |
| 2033 | 3 | 3 | | *4* | *4* | | *4* | *4* | |
| 2109 | 3 | | 3 | *4* | | *4* | *4* | | *4* |
| 2150 | 4 | | | 4 | | | 4 | | |
| 2161 | 1 | | 1 | 1 | | 1 | *3* | | *3* |
| 2232 | | 4 | | | 4 | | | 4 | |
| 2256 | | 3 | | | 3 | | | 3 | |
| 2265 | 3 | | | 3 | | | 3 | | |
| 2269 | | | 4 | | | 4 | | | 4 |
| 2276 | | 4 | 4 | | 4 | 4 | | 4 | 4 |
| 2327 | | | 3 | | | 3 | | | 3 |
| 2351 | 3 | | | *4* | | | *4* | | |
| 2366 | | 3 | 3 | | *4* | *4* | | *4* | 3 |
| 2617 | | 3 | 3 | | 3 | 3 | | 3 | 3 |





**Figure 5**. Daily averaged river discharge and water temperature for the reference (1990 to 2019, solid line) and the far future period (2070 to 2099, dashed line) at 4 stations showing the current and the future thermal hysteresis loops. Flow data used is from model $M_1$, stations belong to the Alpine, Swiss Plateau, Regulated and Downstream Lake thermal regimes. Daily averaged datasets have been smoothed twice with a running average of 30 days. Hysteresis class change in roman numericals (cf. Fig. 4).





## 3.3 Temperature extremes

The analysis is focused on temperature extremes in the summer months (June to August),
during which the severity of extremes varies in between climate scenarios and is different on
individual station basis and on a thermal regime basis (Figure 6). From the reference (1990 to
2019) to the far future (2070 to 2099) period the extreme event severity for scenario RCP2.6
increased on average with +0.20 °C (Figure 6a), by +0.38 °C for RCP4.5 (Figure 6 b) and +0.61
467 °C for RCP8.5 (Figure 6 c).

During the reference period extreme conditions were worst in the Swiss Plateau thermal regime
(mean extreme event severity +2.8 °C) followed by the Downstream Lake (+2.2 °C), Regulated
(+1.3 °C), Alpine (+1.1 °C) and Spring regimes (+0.12 °C). For all climate scenarios and all
thermal regimes, the severity of extreme events increased throughout the 21$^{st}$ century. The
largest increase from the reference to the far future period was found at stations in the Regulated
thermal regime (mean extreme event severity increase RCP2.6: +0.28 °C, RCP4.5: +0.54 °C,
RCP8.5: +0.93 °C) followed by stations in the Swiss Plateau (RCP2.6: +0.26 °C, RCP4.5:
+0.48 °C, RCP8.5: +0.78 °C), Alpine (RCP2.6: +0.23 °C, RCP4.5: +0.45 °C, RCP8.5:
+0.68°C), Downstream Lake (RCP2.6: +0.23 °C, RCP4.5: +0.40 °C, RCP8.5: +0.61 °C) and
Spring regimes (RCP2.6: +0.01 °C, RCP4.5: +0.01 °C, RCP8.5: +0.03 °C). Note that the use
of extreme event severity as an index should be viewed as the minimum temperature increase
of extreme events in the future while it denotes the increase of the 90$^{th}$ percentile.





**Figure 6.** Severity of water temperature extremes from June to August for 30 years of climate simulations (blue bars 1990 to 2019, red bars 2070 to 2099) ordered according to thermal regime. Shown are the lower and upper quartiles (extent of bar) and the median (bar center line) of the difference between the 90th percentile to the seasonal median temperature (30 years of data) from all available climate models (additionally averaged where multiple hydrological models exist) at each station and time period, i.e., the bar extents show climate model induced variability in each period. Stations 2414 and 2462 are not shown since the flow model $M_4$ lacked 30 years of continuous data.





## 3.4 Thermal thresholds

The results presented below represent the number of stations where the daily temperature was
above a given thermal threshold (bar center line Figure 7 above 0). Under the RCP8.5 scenario
from the reference to the far future, the number of stations exceeding the mortality threshold
(25 °C) increased from 4 to 37 stations from a total of 54 stations in the Downstream Lake and
Swiss Plateau regimes (Figure 7a). For the Regulated, Alpine and Spring thermal regime
stations, none passed the lethal threshold during the reference period, but for the far future 1
out of 26 stations exceeded it. For Downstream Lake and Swiss Plateau regime stations, the
PKD threshold (15 °C) was largely exceeded already during the reference period (52 of 54
stations), increasing to all stations in the far future (Figure 7b). For the Regulated, Alpine and
Spring thermal regime stations, 2 out of 26 stations exceeded the PKD threshold already during
the reference period. While in the far future, 20 out of 26 Regulated, Alpine and Spring regime
stations broke through the 15 °C threshold. With respect to fish egg mortality (13 °C) from
September to January, all Downstream Lake regime stations exceeded this threshold both in
the reference period as well as in the far future (Figure 7c). During the reference period, 4 out
of 9 Regulated and 31 out of 34 Swiss Plateau regime stations exceeded the 13 °C threshold.
Correspondingly, for the Regulated and Swiss Plateau regimes, 8 out of 9 and 34 out of 34
stations, respectively, exceeded the 13 °C threshold during the far future period. Although
Alpine regime stations never exceeded the 13 °C threshold during the reference period, 8 out
of 15 stations exceeded this limit during the far future period. From the two groundwater fed
Spring stations, neither the mortality nor the PKD or fish egg mortality thresholds were
exceeded.



**Figure 7.** Number of days superseding thermal threshold for the brown trout for the RCP8.5 climate scenario. a) Mortality threshold at daily mean temperatures >25 °C, b) increased risk for proliferative kidney disease (PKD) at daily mean temperatures >15 °C, egg mortality during September to January at temperatures > 13 °C. Data consist of 30 years of climate simulations (blue bars 1990 to 2019, red bars 2070 to 2099) ordered according to thermal regime. Shown are the median (bar center line) and the lower and upper quartiles (left and right bar extent) of the climate simulation from all available climate models (additionally averaged where multiple hydrological models exist), i.e., the bar extents show climate model induced variability for each period with annual resolution. Stations 2414 and 2462 are not shown since the flow model M₄ lacked 30 years of continuous data.





## 4 Discussion

### 4.1 Multi-fidelity modelling approach

The study of climate change includes the investigation of physical processes on global, regional and local scales. As scales change so too does the required level of detail needed to resolve the different water cycle components that are relevant on the respective scale. An ideally suited approach to address this challenge in hydrological modelling is a multi-fidelity model framework, which combines multiple computational models of varying complexity in an automated selection framework that ensures robust predictions while limiting the computation to only the necessary level of detail (Fernández-Godino, 2023). The use of process dependent fidelity ensures proper representation of physical processes on regional to local scales while keeping computational costs to a minimum. Multi-fidelity modelling is especially useful when acquiring high-accuracy data is costly and/or computationally intensive, as is the case for climate change impact assessment on the hydrological cycle. By combining lower fidelity water temperature models with high-fidelity climate model outputs, in this study we satisfied the vital principle of multi-model analysis that is required for robust climate change impact assessments (Duan et al., 2019).

To expand on previous results of river water temperature projections for Switzerland (Michel et al., 2022), we employed a multi-fidelity modeling approach able to automate the generation of water temperature simulators for the different national river temperature monitoring stations of Switzerland, as summarized in Figure 1. Models of varying complexity were built from integrating high-fidelity climate and hydrological modelling outputs (i.e., downscaled climate (Table 1) and hydrological model outputs (Figure 2a), CH2018 and Hydro-CH2018) with low-fidelity river temperature models of varying degrees of parametrization i.e., air2water and air2stream (Toffolon & Piccolroaz, 2015; Piccolroaz et al., 2013). Statistical learning-based coupling of atmospheric and hydrological stations (Table 2) and classification of river stations into thermal regimes (Figure 2b & 2c) enabled optimal low-fidelity model selection (Figure 2d) and parametrization.

### 4.2 Adjustment of trends

A trend bias correction was applied to the temperature model outputs due to the difference observed between modeled and measured trends (Table B3 to B6). The correction decreased the difference between modeled and measured annual trends by approximately 0.1 °C per decade. After the bias correction, modeled annual trends with climate simulations as inputs followed closely the observed trends (Table B7). Pre-adjustment climate scenarios have a different bias compared to measurements, with RCP8.5 simulations most closely following observed trends while RCP2.6 simulations exhibiting the largest bias. This discrepancy in bias is caused by the averaging of trends from either up to 22 (RCP8.5), 17 (RCP4.5) or 9 (RCP2.6) climate simulations. The trend bias adjustment was applied seasonally, resulting in an adjustment of 0.12 °C per decade on average. The largest adjustment was required for the June to August period (0.22 °C per decade) while the smallest adjustment was made for the December to February period (0.05 °C per decade). Note that only 2 out of 16 Alpine stations had long enough measured datasets (i.e., 30 years) to derive a historical trend, and that trend was used to adjust all 15 stations. The trend adjustment upscaled from 2 to 15 Alpine stations, as well as the calibration at these stations, could thus benefit from longer time series at Alpine stations. We therefore recommend care while using the bias corrected data from the Alpine stations. Additionally, for the groundwater fed station 2499 in the Spring thermal regime, measured water temperature is inversely correlated to air temperature. The result is a near zero or negative trend for the future (below 0 in Figure 4). Although the modeled trend at station



2499 is statistically significant, the result indicates a limitation in the air2stream model to
resolve effectively groundwater dominated processes under climate change.

### 4.3 Warming rates, trends, and hysteresis analysis

As expected and supported by Michel et al., (2020, 2022), the considered climate scenario
turned out to be the most important factor for river water temperature increase, with RCP8.5 at
an average of +0.36 °C per decade warmer river water and +0.49 °C per decade warmer air
temperatures being the scenario that results in the largest warming. The seasonal difference in
the warming of near surface air temperatures observed in Switzerland, with stronger warming
in summer compared to winter (CH2018, 2018), could also be identified in the river water
temperature projections.

Among the different stations, common patterns and trends in river temperature warming could
be identified by classifying the stations into the 4 different river thermal regimes occurring in
Switzerland (Piccolroaz et al., 2016). The classification was further improved in this study by
adding a groundwater spring class and using thermal pattern recognition to regroup river
temperature monitoring stations by automatically identifying key thermal influences from
upstream of a given monitoring station (e.g., the thermal influence of a lake, of tributaries or
of a spring.

In terms of overall warming, the strongest warming on an annual basis emerged for stations in
the Alpine regime, followed, in order, by stations in the Downstream Lake, Regulated, Swiss
Plateau, and Spring regimes (Figure 4). The strong warming of Alpine regime stations has its
origins in the strongest near-surface air temperature warming trend in summer that is occurring
in southern parts of Switzerland (CH2018, 2018). The strong warming in the Downstream Lake
regime can be explained by the extended residence time of water in lakes compared to rivers
in general (allowing longer time for waters to heat up) and to a difference in seasonal patterns,
aspects that the employed air2water model explicitly considers. A coupled river-lake modelling
study in Switzerland (Aare to Lake Biel, Rôhne to Lake Geneva) showed a difference in
epilimnion to river warming rates of + 0.03 to +0.11 °C per decade (Råman Vinnå et al., 2018).

Finally, by using and extending an index developed for classifying hysteretic loops (Zuecco et
al., 2016), it became apparent that climate warming adjust river temperature hysteresis towards
a state with higher temperature and a volume decrease. This is seen as a stretching of most
thermal loops diagonally towards the upper left (Figure 5). The trend stretching results from
the general decrease in discharge as well as the increased seasonal near-surface air temperature
water warming occurring during the summer months. Together, these two processes
predominantly increase water temperature in summer as well.

### 4.4 Thermal extremes

The here proposed "extreme event severity index" together with a removal of the climatic trend
during each period, allowed us to investigate the change in the baseline of extreme temperature
under each thermal regime considered here. The index is independent of past extreme
conditions and relate extremes to the time period being investigated. Like for the water
temperature warming rates and trends, the severity of temperature extremes was impacted the
most by the choice of the climate scenario, similarly so for thermal regimes as a whole and for
individual stations. The largest increase of river temperature extremes occurred under the
RCP8.5 scenario, followed by the RCP4.5 scenario. Noteworthy is that under the RCP2.6
scenario, extreme event frequency and severity stayed more or less constant throughout the 21st
century.



Looking at extreme events at the level of thermal regimes, during the reference period (1990 to 2019), the most sever extreme temperatures occurred at stations in the Swiss Plateau and Downstream Lake regimes. For the far future (2070 to 2099), under all climate scenarios the Swiss Plateau and the Downstream Lake regime stations remain as the stations with the severest extreme events, while the increase in extreme event severity increases the most for the Regulated and the Swiss Plateau regimes. As the Swiss Plateau and Regulated regime stations are mostly located in the Swiss low land in the Northwestern part of Switzerland (see Figure 2b), they are the ones that are expected to experience the most severe low flow conditions, especially in summer months under the RCP8.5 scenario, with a discharge reduction ranging from 5 to 60 % (FOEN, 2021; Brunner, et al., 2019; Brunner, et al., 2019; CH2018, 2018). As the discharge projections have been directly considered in the employed multi-fidelity modelling approach, the strong increase in extreme event severity for these stations is thus a direct result of the expected increased occurrence of low flow events, while the seasonal near-surface air temperature changes are mostly responsible for an increasing median of river water temperatures.

## 4.5 Thermal Thresholds

The likely impact of climate change under the RCP8.5 scenario was investigated with known thermal thresholds for the brown trout (i.e., risk of death at 25 °C and above; increased occurrence of PKD above 15 °C; increased fish egg mortality at 13 °C between September and January), a cold water fish species that is found in rivers and streams throughout all of Switzerland (Brodersen et al., 2023). While brown trout's can in principle die already after about 10 min at temperatures of 30 °C (Elliott, 1981), due to the daily temporal resolution of the employed models, thermal thresholds were only evaluated on a daily time scale. Even when looking only at the daily time scale, the results of this study are cause for concern, as both the number of stations as well as the duration during which thermal thresholds are exceeded increase. Viewed alongside the fact that the number of catches of brown trout in Switzerland have already severely decreased in the past decades, for example from 73,500 in 1989 to 12,750 in 2019 in the rivers of the Swiss canton of Bern, which represents rivers of all types of thermal regimes that are found in Switzerland (FOEN, 2024), the outlook for the brown trout's future in Swiss rivers is grim. Our results show clear thermal regime dependent differences for the present and future thermal related stress on the brown trout (Figure 7). The lethal threshold (25°C) was seldomly exceeded in the past (Figure 7a). However, towards the end of the 21st century, for a majority of stations in the Downstream Lake and Swiss Plateau thermal regimes the lethal threshold was exceeded on at least one day during the year, making areas which could previously be considered safe for the brown trout potentially lethal at least on certain days of the year. In addition, the 25 °C limit is also critical for anthropogenic water use in Switzerland, as the Swiss law (Water Protection Ordinance 814.201) prohibits a thermal use of waters for cooling purposes beyond this threshold. Unfortunately, our results not only show an increased occurrence of lethal temperatures, but also the less imminently lethal but nevertheless detrimental lower temperature threshold of the increased occurrence of the PKD disease (15 °C) will be exceeded much more frequently (see Figure 7b), as will the threshold for fish egg mortality (Figure 7c). Alpine stations, and to a lesser extend Regulated stations, where previously the thermal conditions for an increased likelihood of PKD were not met, are likely also going to exhibit these conditions in the warmer summer months. Given the 153 days from September to January, egg development (approx. 30 to 90 days Alp et al., 2010) should still have enough time to take place safely throughout the 21st century in Regulated, Swiss Plateau, Alpine and Spring thermal regime rivers. Rivers in the Downstream Lake thermal regime are likely too large to facilitate spawning and were therefore not further considered in this analysis.



The thermal analyses preformed here do not resolve all the processes affecting fishes'
sensitivities to thermal extremes or spawning success. The ability to migrate, find local cold
water refugia, or the availability for bottom gravel substrate required for spawning was not
explicitly simulated. However, as severe temperature extremes which exceed the fish mortality
threshold of 25°C can in general occur in tandem with low flow conditions (see Figure 5), the
possibilities for the brown trout to temporally migrate to a cold water refugia during such
extremes can be expected to be strongly limited. And while we did not investigate the
temperature to initiate spawning, it is likely that longer occurrence of high water temperature
periods during Autumn will have the potential to delay brown trout spawning. Moreover, due
to increased river discharge and erosion in winter, sufficient bottom gravel substrate for
spawning can be expected to decrease in future (Junker et al., 2015). Hence, to conclude, a
changing climate will significantly increase the stress on brown trout, and given the widespread
distribution of this fish species, future changes in temperature related death of adults cause us
most concern.

## 5. Summary and Conclusions

An automated multi-fidelity modelling approach consisting of downscaled regional climate
models, hydrological catchment models, and two semi-empirical water temperature models at
variable degrees of parametrization complexity was used to investigate future river water
temperatures across Switzerland under three climate scenarios. Model selection and
performance was optimized by grouping catchments under thermal regimes using a process
consisting of thermal pattern recognition with hierarchical clusters.

According to the simulations, for the high emission climate scenario (RCP8.5), average river
water temperatures across Switzerland will increase by 3.0 °C (0.37 °C per decade from 1990
to 2099), while under the low emission scenario (RCP2.6) temperatures increase by only 0.9
677 °C. The strongest river water warming under the high emission scenario can be expected to
678 occur in the Alpine thermal regime (+3.5 °C) followed by stations in the Downstream Lake
regime (+3.4 °C). A general shift in river discharge with less water in summer and more water
in winter together with increased warming in summer produced increased seasonal warming
which stretched hysteresis loops of water temperature versus discharge. The severity of thermal
extremes in summer increased by, on average, 0.6 °C under the high emission scenario, while
under the low emission scenario the increase was limited to 0.2 °C. Caused by future low flows,
rivers stations in the Swiss Plateau thermal regime showed the most severe absolute river
temperature extremes during the reference period, while the absolute extreme temperature
change was largest in Regulated thermal regime stations (RCP2.6: +0.28 °C, RCP4.5: +0.54
687 °C, RCP8.5: +0.93 °C). Our results show increased future thermal stress on cold-water fishes
such as the brown trout, with substantial increases in the duration of threshold exceeding
temperatures. These exceedances will lead to the increased likelihood of reproduction
difficulties, occurrence of sickness and high temperature related mortality for brown trout in
rivers where this previously was not a problem.

A multi-fidelity modelling approach was deemed necessary to work around computational
limitations while investigating regional climate change across Switzerland. We show how
surface water temperature models can be employed for various different thermal regimes by
automatically adapting their parametrization complexity to the required level, including for
stations downstream of lakes that are influenced strongly by the lake thermal regimes. Yet,
future studies would benefit from connecting lakes and rivers in one modelling framework.
The climate models used here were part of to the global CMIP5 and regional EUROCORDEX



coordinated modeling efforts (CH2018, 2018). Future studies should however consider using
the more recent CMIP6 or later collaborations for their projections.

Swiss water protection management leans on the sensitivity of species for enforcing thermal
utility rules prohibiting thermal use past certain thresholds (Waters Protection Ordinance
814.201). Our results show a change in the duration and the location of threshold exceeding
water temperatures, which threatens not only the brown trout but have implications for future
anthropogenic use of Swiss surface waters. Local and regional climate protection measures to
limit negative effects of climate change includes but are not limited to the creation of river
bank shading (Trimmel et al., 2018), dam management (Payne et al., 2004), river restoration,
stormwater and site-specific management (Palmer et al., 2008) as well as managed ground
water recharge (Epting et al., 2023). Ultimately in the work to mitigate negative climate impact,
management needs to weight the need for protection and preservation with its associated cost
and benefit towards the outcome of a non-interactive, partial or full climate protection
approach.



## Data availability

Atmospheric temperature climate data from the CH2018 project was obtained from the Swiss National Centre for Climate Services (nccs.admin.ch) data portal. On the same portal, discharge datasets from the Hydro-CH2018 project are available but at a temporally limited scale (monthly, seasonally and yearly means). We required daily resolved discharge data which was obtained directly from Massimiliano Zappa (model M1), Daphné Freudiger (M3), and Adrien Michel (M4). Data from model M2 (Muelchi et al., 2021) is available at http://doi.org/10.5281/zenodo.3937485. All river water temperature model results for climate models analyzed and left out (Table 1) and adjusted datasets of air temperature and discharge produced here will be made publicly available upon publishing of this work.

## Author contributions

LRV and JE came up with the concept and secured the funding. VB designed and performed the thermal pattern recognition, VB and LRV implemented it for ordering catchments according to thermal regimes. LRV conducted the forcing data adjustment, model setup and use. LRV and JE conducted the analysis of the results. OS provided scientific support. All authors took part in the writing of this manuscript.

## Acknowledgments

We acknowledge the support from Martin Schmid for external scientific quality control, Amber van Hamel for valuable insights in the thermal extreme analysis, Sebastiano Piccolroaz for guidance in the use of the air2stream and the air2water models, Thilo Herold at the Swiss Federal Office of the Environment (FOEN) and the Freiwillige Akademische Gesellschaft (FAG) Basel for funding this work.

## Competing interests

The authors declare no competing interests.



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



# Appendix A: Description of water temperature models

**air2stream**

The river temperature model air2stream can be used with five different degrees of complexity, which differ in their level of parameterization (Piccolroaz et al., 2016; Toffolon & Piccolroaz, 2015), where some parameters are neglected (Eq. 1 to 5). In air2stream, water temperature ($T_w$) [°C] is calculated from air temperature ($T_a$) [°C] and from discharge ($Q$) in either a 3-, 4-, 7-, or 8-parameter configuration.

*8-parameter version*

$$\frac{\Delta T_w}{\Delta t} = \frac{1}{\delta}\left\{a_1 + a_2 T_a(t) - a_3 T_w(t) + \theta\left[a_5 + a_6 \cos\left(2\pi\left(\frac{t}{t_y} - a_7\right)\right) - a_8 T_w(t)\right]\right\} \tag{1}$$

where, $T_w$ is water temperature, $T_a$ air temperature, $t$ represents the day of the year, $t_y$ is the duration of one year, $a_1$ is a fitting parameter with units °C/day and $a_2$-$a_8$ are dimensionless fitting parameters, $\delta$ represents the dimensionless depth and is defined as $\delta = \theta^{a_4}$, while $\theta$ represents the dimensionless flow defined as $\theta = Q(t)/\bar{Q}$, with $Q(t)$ being flow and $\bar{Q}$ the mean flow.

*7-parameter version:*

$$\frac{\Delta T_w}{\Delta t} = a_1 + a_2 T_a(t) - a_3 T_w(t) + \theta\left[a_5 + a_6 \cos\left(2\pi\left(\frac{t}{t_y} - a_7\right)\right) - a_8 T_w(t)\right] \tag{2}$$

Here, $\delta$ is set equal to 1 and the influence of river depth on water temperature is not explicitly considered anymore.

*5-parameter version:*

$$\frac{\Delta T_w}{\Delta t} = a_1 + a_2 T_a(t) - a_3 T_w(t) + a_6 \cos\left(2\pi\left(\frac{t}{t_y} - a_7\right)\right) \tag{3}$$

With both $\delta$ and $\theta$ set to 1, no depth or discharge input is required and the effect of both depth and discharge on water temperature is approximated by the fitting constant $a_1$.

The 3- and 4-parameter versions are recommended for cases where both discharge and the thermal effect of tributaries at a given observation point along a stream are considered small.

*4-parameter version:*

$$\frac{\Delta T_w}{\Delta t} = \frac{1}{\delta}\{a_1 + a_2 T_a(t) - a_3 T_w(t)\} \tag{4}$$

In this version, $\theta$ is set to 0 and it is assumed that the mean temperature of tributaries is approximately equal to the temperature of the river itself, i.e., the longitudinal (spatial) gradient of temperature is small. Moreover, seasonal effects are neglected.

*3-parameter version:*

$$\frac{\Delta T_w}{\Delta t} = a_1 + a_2 T_a(t) - a_3 T_w(t) \tag{5}$$

In this simplest version of air2stream, $\theta$ is set to 0 and $\delta$ to 1, such that no discharge input is required and flow, depth, seasonality, and temperature gradients are approximated via fitting the constant $a_1$.



**air2water**

With the air2water model, surface water temperature ($T_w$) [°C] is calculated towards a reference
temperature ($T_r$) [°C], with air temperature ($T_a$) [°C] as the only input. $T_r$ links surface temperature to
bottom temperature. The lake model can be used in three versions (Piccolroaz, 2016; Toffolon et al.,
2014; Piccolroaz et al., 2013), with 8, 6, or 4 parameters (Eq. 6 to 8).

*8-parameter version*

$$\frac{\Delta T_w}{\Delta t} = \frac{1}{\delta}\left\{a_1 + a_2 T_a - a_3 T_w + a_5 \cos\left[2\pi\left(\frac{t}{t_y} - a_6\right)\right]\right\}$$
(6)

In the 8-parameter version all dimensionless fitting parameters $a_1$-$a_8$ are active together with δ known
as the volume ratio or normalized depth defined as:

$$\delta = exp\left(-\frac{T_w - T_r}{a_4}\right) \qquad \text{for } (T_w \geq T_r)$$

$$\delta = exp\left(-\frac{T_r - T_w}{a_7}\right) + exp\left(-\frac{T_w}{a_8}\right) \quad \text{for } (T_w < T_r)$$

δ is theoretically defined in the range between 0 and 1, with the value 1 corresponding to the maximum
volume of the surface layer, decreasing values account for increasingly strong stratification, which
reduce the water volume affected by the surface heat budget (Toffolon et al., 2014). $T_w < T_r$ represent a
inversely stratified lake in winter with colder water (< 4 °C) on-top of warmer, while $T_w > T_r$ represent
a stratified lake in summer with warmer water (> 4 °C) on top of colder water (Piccolroaz et al., 2013).
Ice is not included in the model.

*6-parameter version;*

$$\frac{\Delta T_w}{\Delta t} = \frac{1}{\delta}\left\{a_1 + a_2 T_a - a_3 T_w + a_5 \cos\left[2\pi\left(\frac{t}{t_y} - a_6\right)\right]\right\}$$
(7)

$$\delta = exp\left(-\frac{T_w - T_r}{a_4}\right) \qquad \text{for } (T_w \geq T_r)$$

$$\delta = 1 \qquad \text{for } (T_w < T_r)$$

In the 6-parameter version, δ is set to 1 for $T_w < T_r$ i.e., the lake does not become inversely stratified.

*4-parameter version*

$$\frac{\Delta T_w}{\Delta t} = \frac{1}{\delta}\{a_1 + a_2 T_a - a_3 T_w\}$$
(8)

$$\delta = exp\left(-\frac{T_w - T_r}{a_4}\right) \qquad \text{for } (T_w \geq T_r)$$

$$\delta = 1 \qquad \text{for } (T_w < T_r)$$

Here, $a_5$ is set to 0 and, as in the 6-parameter version, δ is set 1 for $T_w < T_r$. By setting $a_5$ to 0, the 4-
parameter version lacks the imposed sinusoidal forcing. Additionally, the physical meaning of
parameters differs here from the 8-parameter version, as the terms including $T_a$ and $T_w$ now indirectly
consider the periodicity of external meteorological forcing's. This version is preferable when the annual
cycles of $T_a$ or $T_w$ are approximately sinusoidal (Piccolroaz, 2016).



## Appendix B: Supporting Figures and Tables

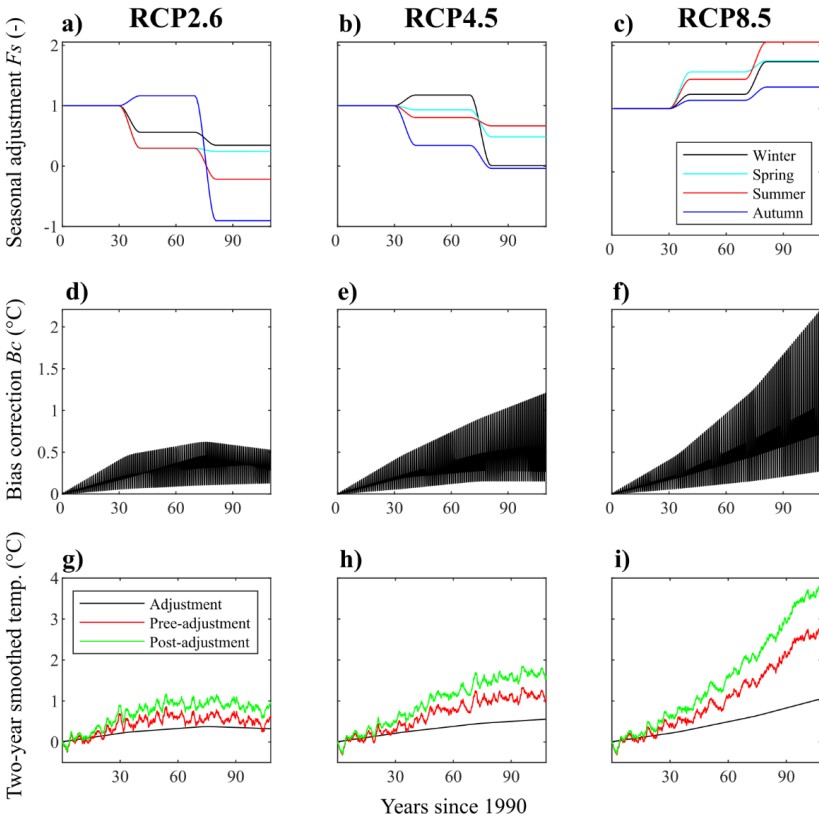

**Figure B1.** Trend bias correction example for station 2612 belonging to the Swiss Plateau regime simulated with air2stream. (a-c): seasonal adjustment factors for winter (December to February), spring (March-May), summer (June-August), autumn (September-November). (d-f) seasonal and thermal regime dependent bias correction $Bc$. (g-i) $Bc$ added to the projections of river temperature.





**Table B1.** Temperature model calibration setup and cluster results. ALP: Alpine regime; DLA: Downstream lake
regime; SPJ: Swiss Plateau regime; HYP: Influenced by hydropeaking; 3/5*: discharge data not available therefore
only air2stream tested with 3 and 5 parameters.

| ID | Tested model/s | Thermal regime | | | Thermal clusters |
|----|----------------|----------------|--|--|------------------|
| | | **Derived here** | Michel et al., 2020 | Piccolroaz et al., 2016 | **DTWARP_PER_33** |
| 2009 | air2stream | Regulated | HYP | Regulated | Cluster 7.3 |
| 2016 | air2stream & air2water | Downstream Lake | DLA | Outlet | Cluster 2.4 |
| 2018 | air2stream & air2water | Downstream Lake | DLA | | Cluster 2.4 |
| 2019 | air2stream3/5* | Regulated | HYP | Regulated | Cluster 8.1 |
| 2029 | air2stream & air2water | Downstream Lake | DLA | Outlet | Cluster 2.4 |
| 2030 | air2stream & air2water | Downstream Lake | DLA | Outlet | Cluster 3.2 |
| 2033 | air2stream | Alpine | | | Cluster 8.4 |
| 2034 | air2stream | Swiss Plateau | SPJ | Natural low-land | Cluster 3.3 |
| 2044 | air2stream | Swiss Plateau | SPJ | Natural low-land | Cluster 3.3 |
| 2056 | air2stream | Regulated | HYP | Regulated | Cluster 8.2 |
| 2068 | air2stream3/5* | Regulated | HYP | | Cluster 6 |
| 2070 | air2stream | Swiss Plateau | SPJ | Natural low-land | Cluster 6 |
| 2084 | air2stream | Regulated | HYP | | Cluster 7.3 |
| 2085 | air2stream & air2water | Downstream Lake | DLA | Outlet | Cluster 2.3 |
| 2091 | air2stream & air2water | Downstream Lake | DLA | Outlet | Cluster 1.3 |
| 2104 | air2stream & air2water | Downstream Lake | DLA | | Cluster 3.2 |
| 2106 | air2stream | Swiss Plateau | SPJ | | Cluster 3.6 |
| 2109 | air2stream | Alpine | ALP | | Cluster 8.2 |
| 2112 | air2stream | Swiss Plateau | | Natural low-land | Cluster 7.2 |
| 2113 | air2stream3/5* & air2water | Downstream Lake | | | Cluster 1.3 |
| 2126 | air2stream | Swiss Plateau | | Natural low-land | Cluster 3.5 |
| 2130 | air2stream3/5* & air2water | Downstream Lake | | | Cluster 1.3 |
| 2135 | air2stream & air2water | Downstream Lake | DLA | Outlet | Cluster 3.2 |
| 2139 | air2stream3/5* | Swiss Plateau | | | |
| 2143 | air2stream & air2water | Downstream Lake | DLA | Outlet | Cluster 2.2 |
| 2150 | air2stream | Alpine | | | Cluster 7.2 |
| 2152 | air2stream & air2water | Downstream Lake | DLA | Outlet | Cluster 2.4 |
| 2159 | air2stream | Swiss Plateau | | Natural low-land | Cluster 4.3 |
| 2161 | air2stream | Alpine | | Snow-fed | Cluster 10 |
| 2167 | air2stream & air2water | Downstream Lake | | | Cluster 1.1 |
| 2170 | air2stream3/5* | Swiss Plateau | ALP | | Cluster 6 |
| 2174 | air2stream3/5* & air2water | Downstream Lake | DLA | | Cluster 2.3 |
| 2176 | air2stream | Swiss Plateau | | | |
| 2179 | air2stream | Swiss Plateau | | Natural low-land | Cluster 5.2 |
| 2181 | air2stream | Swiss Plateau | | | |
| 2210 | air2stream | Swiss Plateau | | | Cluster 4.3 |
| 2232 | air2stream | Alpine | | Snow-fed | Cluster 8.4 |
| 2243 | air2stream & air2water | Downstream Lake | DLA | | Cluster 1.3 |
| 2256 | air2stream | Alpine | | Snow-fed | Cluster 9 |
| 2265 | air2stream | Alpine | | | |
| 2269 | air2stream | Alpine | ALP | | Cluster 9 |
| 2276 | air2stream | Alpine | | Snow-fed | Cluster 7.3 |
| 2282 | air2stream3/5* | Swiss Plateau | | | Cluster 7.2 |
| 2288 | air2stream & air2water | Downstream Lake | | | Cluster 2.2 |
| 2290 | air2stream3/5* | Spring | | | |
| 2307 | air2stream | Swiss Plateau | | | Cluster 6 |
| 2308 | air2stream | Swiss Plateau | | Natural low-land | Cluster 5.2 |
| 2327 | air2stream | Alpine | | Snow-fed | Cluster 9 |
| 2343 | air2stream | Swiss Plateau | | Natural low-land | Cluster 5.3 |
| 2347 | air2stream3/5* | Alpine | | Natural low-land | Cluster 8.3 |
| 2351 | air2stream | Alpine | | | Cluster 8.2 |
| 2366 | air2stream | Alpine | | Snow-fed | Cluster 9 |
| 2369 | air2stream | Swiss Plateau | | Natural low-land | Cluster 5.2 |
| 2372 | air2stream | Regulated | HYP | Regulated | Cluster 7.3 |
| 2374 | air2stream | Swiss Plateau | | Natural low-land | Cluster 6 |
| 2386 | air2stream | Swiss Plateau | | | Cluster 3.1 |
| 2392 | air2stream3/5* & air2water | Downstream Lake | | | Cluster 2.2 |
| 2410 | air2stream3/5* | Swiss Plateau | | | Cluster 5.4 |
| 2414 | air2stream | Swiss Plateau | | Natural low-land | Cluster 6 |
| 2415 | air2stream | Swiss Plateau | SPJ | Natural low-land | Cluster 1.3 |
| 2432 | air2stream | Swiss Plateau | | | Cluster 3.5 |
| 2433 | air2stream3/5* | Swiss Plateau | | | Cluster 6 |
| 2434 | air2stream | Swiss Plateau | | | |
| 2457 | air2stream3/5* & air2water | Downstream Lake | DLA | Snow-fed | Cluster 4.4 |
| 2462 | air2stream | Alpine | ALP | | Cluster 9 |
| 2467 | air2stream3/5* | Regulated | | | Cluster 5.1 |
| 2473 | air2stream | Regulated | HYP | | Cluster 6 |
| 2481 | air2stream | Regulated | HYP | | Cluster 7.3 |
| 2485 | air2stream3/5* | Swiss Plateau | | | Cluster 3.4 |
| 2493 | air2stream | Swiss Plateau | | | Cluster 5.3 |
| 2499 | air2stream3/5* | Spring | | | |
| 2500 | air2stream | Swiss Plateau | SPJ | | Cluster 4.4 |
| 2604 | air2stream | Swiss Plateau | | | Cluster 7.1 |
| 2606 | air2stream & air2water | Downstream Lake | | | Cluster 1.3 |
| 2608 | air2stream3/5* | Swiss Plateau | | Natural low-land | Cluster 5.1 |
| 2609 | air2stream | Swiss Plateau | | Natural low-land | Cluster 6 |
| 2612 | air2stream | Swiss Plateau | | Natural low-land | Cluster 7.1 |
| 2613 | air2stream3/5* & air2water | Downstream Lake | | | Cluster 1.3 |
| 2617 | air2stream | Alpine | | Snow-fed | Cluster 8.2 |
| 2623 | air2stream3/5* | Alpine | | | Cluster 9 |
| 2634 | air2stream | Swiss Plateau | SPJ | | |
| 2635 | air2stream3/5* | Swiss Plateau | | | |





**Table B2.** Best performing model setup using air2stream (TM1) and air2water (TM2), with corresponding calibration parameter limits (see Table 5).

| Stations Air-River | Model | Calibration | | | Validation | | Parameter Values | | | | | | | |
|---|---|---|---|---|---|---|---|---|---|---|---|---|---|---|
| | | Time | RMSE (°C) | Mean Q (m³ s⁻¹) | Time | RMSE (°C) | $a_1$ | $a_2$ | $a_3$ | $a_4$ | $a_5$ | $a_6$ | $a_7$ | $a_8$ |
| AIG-2009 | TM1 | 1990-2019 | 0.52 | 184.74 | 1981-1989 | 0.59 | -0.057 | 0.362 | 0.183 | 0.185 | 12.158 | 3.850 | 0.533 | 1.921 |
| BUS-2016 | TM1 | 1990-2019 | 0.81 | 309.96 | 1985-1989 | 0.98 | 0.603 | 0.180 | 0.156 | | 3.849 | 2.325 | 0.603 | 0.357 |
| BUS-2018 | TM2 | 1990-2019 | 0.96 | | 1985-1989 | 1.18 | 1.137 | 0.090 | 0.169 | 9.939 | 0.549 | 0.626 | | |
| MER-2019 | TM1 | 1990-2019 | 0.85 | 36.53 | 1980-1989 | 0.68 | 5.044 | 0.273 | 1.233 | | | | | |
| BER-2029 | TM2 | 1990-2019 | 0.87 | | 1980-1989 | 0.93 | 0.181 | 0.023 | 0.032 | 12.592 | 0.052 | 0.614 | | |
| INT-2030 | TM1 | 1990-2017 | 0.95 | | 1980-1989 | 1.05 | 0.398 | 0.022 | 0.054 | 5.819 | 0.156 | 0.663 | | |
| CHU-2033 | TM1 | 2002-2019 | 0.75 | 30.91 | | | 0.407 | 0.364 | 0.496 | -0.690 | 8.233 | 5.276 | 0.585 | 1.406 |
| PAY-2034 | TM1 | 1990-2019 | 0.78 | 7.51 | 1980-1989 | 0.84 | 1.736 | 0.749 | 0.748 | | 6.549 | 3.759 | 0.579 | 0.719 |
| SHA-2044 | TM1 | 1990-2019 | 0.80 | 46.37 | 1982-1989 | 0.78 | 1.848 | 0.506 | 0.537 | | 4.394 | 2.759 | 0.582 | 0.519 |
| ALT-2056 | TM1 | 1990-2019 | 0.58 | 42.68 | 1980-1989 | 0.76 | 8.725 | 1.265 | 2.981 | -0.996 | 9.003 | 8.097 | 0.613 | 1.628 |
| MAG-2068 | TM1 | 1997-2019 | 1.04 | 72.29 | 1980-1982 | 0.99 | 0.376 | 0.046 | 0.101 | | | | | |
| LAG-2070 | TM1 | 1990-2019 | 0.85 | 11.74 | 1980-1989 | 1.07 | 3.984 | 0.563 | 0.880 | | 5.420 | 4.985 | 0.586 | 0.780 |
| ALT-2084 | TM1 | 1990-2019 | 0.78 | 19.19 | 1980-1989 | 0.88 | 1.118 | 0.609 | 0.638 | -0.805 | 18.147 | 4.980 | 0.599 | 2.744 |
| BER-2085 | TM1 | 1990-2016 | 0.84 | 175.20 | 1984-1989 | 1.05 | 1.488 | 0.144 | 0.158 | | 2.848 | 2.157 | 0.606 | 0.322 |
| BAS-2091 | TM2 | 1990-2007 | 0.84 | | 1980-1989 | 0.97 | 0.308 | 0.034 | 0.055 | 12.167 | 0.131 | 0.600 | | |
| GLA-2104 | TM2 | 1990-2019 | 1.14 | | 1980-1989 | 1.17 | 0.053 | 0.010 | 0.013 | 6.323 | | | | |
| BAS-2106 | TM1 | 1990-2019 | 0.60 | 15.47 | 1980-1989 | 0.69 | 0.649 | 0.359 | 0.375 | | 6.512 | 1.815 | 0.574 | 0.664 |
| INT-2109 | TM1 | 1990-2019 | 0.60 | 19.08 | 1980-1989 | 0.74 | 9.036 | 1.678 | 3.578 | -0.001 | 29.278 | 4.740 | 0.476 | 5.000 |
| STG-2112 | TM1 | 2006-2019 | 0.80 | 3.16 | | | -0.417 | 0.344 | 0.316 | | 9.488 | 4.955 | 0.581 | 1.299 |
| BUS-2113 | TM2 | 1990-2019 | 0.99 | | 1985-1989 | 1.19 | 0.468 | 0.041 | 0.067 | 11.136 | 0.154 | 0.641 | | |
| TAE-2126 | TM1 | 2002-2019 | 0.61 | 1.70 | | | 4.576 | 0.486 | 0.719 | -0.045 | 9.981 | 5.483 | 0.596 | 1.072 |
| RUE-2130 | TM2 | 1983-1985 | 0.78 | | | | 0.378 | 0.030 | 0.054 | 10.808 | 0.160 | 0.602 | | |
| BER-2135 | TM2 | 1990-2019 | 0.91 | | 1980-1989 | 1.13 | 0.489 | 0.026 | 0.066 | 4.473 | 0.199 | 0.651 | | |
| VAD-2139 | TM1 | 2016-2017 | 0.70 | 10.70 | | | 8.749 | 0.296 | 1.112 | | 2.818 | 0.586 | | |
| KLO-2143 | TM2 | 1990-2017 | 0.91 | | 1980-1989 | 1.05 | 0.185 | 0.033 | 0.042 | 12.721 | 0.057 | 0.628 | | |
| RAG-2150 | TM1 | 2003-2019 | 0.75 | 21.61 | | | 2.292 | 0.592 | 1.058 | -0.813 | 4.882 | 5.460 | 0.580 | 0.937 |
| LUZ-2152 | TM2 | 1990-2019 | 0.94 | | 1980-1989 | 1.22 | 0.254 | 0.023 | 0.040 | 6.979 | 0.105 | 0.632 | | |
| BER-2159 | TM1 | 2007-2019 | 0.71 | 2.57 | | | 4.870 | 1.025 | 1.292 | 0.101 | 14.207 | 7.657 | 0.585 | 1.518 |
| GRC-2161 | TM1 | 2003-2019 | 0.27 | 15.76 | | | 0.987 | 0.164 | 1.343 | -0.054 | 5.115 | 1.352 | 0.356 | 5.000 |
| LUG-2167 | TM2 | 2003-2017 | 0.81 | | | | 0.162 | 0.028 | 0.036 | 9.833 | 0.091 | 0.612 | | |
| GVE-2170 | TM1 | 1990-2017 | 0.93 | 72.30 | 1980-1989 | 0.79 | 15.000 | 0.833 | 2.856 | | 3.416 | 0.568 | | |
| GVE-2174 | TM2 | 1990-2017 | 1.49 | | 1980-1989 | 1.57 | 0.680 | 0.054 | 0.107 | 5.359 | 0.268 | 0.666 | | |
| SMA-2176 | TM1 | 1990-2019 | 0.93 | 6.76 | 1986-1989 | 1.07 | 0.219 | 0.611 | 0.476 | | 6.710 | 4.764 | 0.556 | 0.779 |
| BER-2179 | TM1 | 2004-2019 | 0.81 | 8.16 | | | 1.182 | 0.554 | 0.618 | | 5.696 | 4.287 | 0.585 | 0.672 |
| GUT-2181 | TM1 | 2014-2019 | 0.75 | 35.02 | 1980-1989 | 0.95 | 0.281 | 0.584 | 0.515 | 0.111 | 5.129 | 2.628 | 0.575 | 0.614 |
| FAH-2210 | TM1 | 2002-2019 | 0.86 | 30.66 | | | -0.351 | 0.268 | 0.177 | | 10.405 | 4.313 | 0.557 | 1.062 |
| ABO-2232 | TM1 | 2002-2017 | 0.71 | 1.21 | | | 0.739 | 0.274 | 0.376 | | 5.840 | 4.383 | 0.576 | 1.130 |
| REH-2243 | TM2 | 1990-2019 | 1.06 | | 1980-1989 | 1.22 | 0.276 | 0.029 | 0.044 | 9.131 | 0.126 | 0.623 | | |
| SAM-2256 | TM1 | 2004-2019 | 0.64 | 2.82 | | | 3.067 | 1.065 | 1.959 | | 14.254 | 9.658 | 0.571 | 3.247 |
| SCU-2265 | TM1 | 2016-2019 | 0.67 | 19.28 | | | 1.396 | 0.572 | 0.748 | | 3.617 | 5.290 | 0.570 | 0.981 |
| GRC-2269 | TM1 | 1990-2019 | 0.64 | 4.72 | 1980-1989 | 1.24 | 7.568 | 0.783 | 3.173 | -0.526 | 15.702 | 10.000 | 0.597 | 3.887 |
| ALT-2276 | TM1 | 2005-2019 | 0.65 | 1.80 | | | 3.925 | 0.165 | 0.751 | -2.931 | | | | |
| NAP-2282 | TM1 | 2002-2017 | 0.82 | 16.18 | | | 2.234 | 0.205 | 0.493 | | 1.602 | 0.576 | | |
| SHA-2288 | TM2 | 2009-2017 | 0.85 | | | | 0.117 | 0.028 | 0.033 | 14.258 | 0.031 | 0.659 | | |
| BRL-2290 | TM1 | 2010-2012 | 0.29 | 4.17 | | | 1.935 | 0.017 | 0.265 | | | | | |
| CHA-2307 | TM1 | 2005-2019 | 0.71 | 4.08 | | | -0.877 | 0.135 | - | | 15.923 | 3.325 | 0.553 | 1.845 |
| GUT-2308 | TM1 | 2005-2019 | 0.83 | 1.36 | | | 0.101 | 0.619 | 0.639 | 0.170 | 3.602 | 2.236 | 0.591 | 0.432 |
| DAV-2327 | TM1 | 2004-2019 | 0.64 | 1.72 | | | 11.627 | 1.473 | 3.376 | | 1.574 | 13.805 | 0.586 | 1.979 |
| WYN-2343 | TM1 | 2002-2019 | 0.51 | 1.17 | | | 8.665 | 1.078 | 2.012 | -0.430 | 10.759 | 6.667 | 0.620 | 1.141 |
| GRO-2347 | TM1 | 2003-2017 | 0.96 | 0.45 | | | 1.410 | 0.438 | 1.175 | | 3.711 | 0.639 | | |
| VIS-2351 | TM1 | 2003-2019 | 0.67 | 16.62 | | | -1.184 | 0.362 | 0.210 | | 17.902 | 7.284 | 0.582 | 3.072 |
| BEH-2366 | TM1 | 2011-2019 | 0.83 | 0.55 | | | 0.496 | 0.053 | 0.141 | 0.498 | | | | |
| PAY-2369 | TM1 | 2002-2019 | 0.75 | 1.43 | | | 0.612 | 0.611 | 0.707 | | 4.272 | 2.629 | 0.591 | 0.423 |
| GLA-2372 | TM1 | 1990-2019 | 0.49 | 31.97 | 1980-1989 | 0.64 | 10.472 | 0.843 | 2.093 | -0.562 | 13.870 | 7.625 | 0.608 | 2.140 |
| EBK-2374 | TM1 | 2007-2019 | 0.74 | 3.16 | | | 1.268 | 0.744 | 0.780 | 0.193 | 6.815 | 3.868 | 0.592 | 0.884 |
| TAE-2386 | TM1 | 2007-2019 | 0.64 | 3.63 | | | 2.318 | 0.573 | 0.633 | | 9.231 | 4.664 | 0.579 | 0.921 |
| SHA-2392 | TM2 | 1990-2019 | 0.90 | | 1982-1989 | 0.95 | 0.127 | 0.027 | 0.033 | 12.930 | 0.042 | 0.627 | | |
| VAD-2410 | TM1 | 1996-2017 | 0.57 | 4.82 | | | 12.705 | 0.274 | 1.661 | | 1.971 | 0.622 | | |
| EBK-2414 | TM1 | 2002-2019 | 0.66 | 97.72 | | | 1.022 | 0.485 | 0.653 | | 7.185 | 4.049 | 0.622 | 0.756 |
| KLO-2415 | TM1 | 1990-2019 | 0.69 | 7.84 | 1980-1989 | 0.84 | 4.738 | 0.578 | 0.759 | 0.209 | 9.446 | 6.190 | 0.588 | 0.804 |
| PUY-2432 | TM1 | 2002-2019 | 0.64 | 3.61 | | | 0.896 | 0.426 | 0.483 | | 5.261 | 1.902 | 0.585 | 0.553 |
| CGI-2433 | TM1 | 2011-2019 | 1.53 | 4.94 | | | 1.761 | 0.244 | 0.489 | | 0.467 | 0.935 | | |
| WYN-2434 | TM1 | 2014-2019 | 0.65 | 2.78 | | | 1.999 | 0.768 | 0.812 | 0.278 | 9.241 | 2.520 | 0.575 | 0.954 |
| INT-2457 | TM2 | 1990-2003 | 1.09 | | 1980-1989 | 1.21 | 0.093 | 0.010 | 0.018 | 5.727 | | | | |
| SAM-2462 | TM1 | 1999-2019 | 0.68 | 21.11 | | | 4.733 | 0.705 | 0.968 | | 11.342 | 11.816 | 0.574 | 2.740 |
| BER-2467 | TM1 | 2004-2019 | 0.80 | 49.02 | | | 0.128 | 0.032 | 0.043 | | | | | |
| VAD-2473 | TM1 | 1990-2019 | 0.68 | 230.82 | 1980-1989 | 0.82 | 1.107 | 0.257 | 0.287 | | 5.955 | 3.865 | 0.568 | 0.914 |
| LUZ-2481 | TM1 | 1990-2019 | 0.44 | 12.31 | 1983-1989 | 0.50 | 7.429 | 0.929 | 2.164 | -0.210 | 15.747 | 4.281 | 0.617 | 2.270 |
| FAH-2485 | TM1 | 2002-2019 | 1.10 | 3.06 | | | 6.004 | 0.271 | 0.735 | | 1.127 | 0.552 | | |
| CGI-2493 | TM1 | 2012-2019 | 0.61 | 1.63 | | | 1.936 | 0.454 | 0.658 | -0.354 | 12.248 | 3.157 | 0.633 | 1.410 |
| ALT-2499 | TM1 | 2009-2019 | 0.19 | 1.85 | | | 2.903 | - | 0.457 | | | | | |
| BER-2500 | TM1 | 1990-2019 | 0.61 | 1.01 | | | 4.626 | 0.664 | 1.006 | 0.383 | 12.431 | 5.116 | 0.596 | 1.241 |
| EIN-2604 | TM1 | 2003-2019 | 0.89 | 1.07 | | | 0.565 | 0.546 | 0.584 | | 5.292 | 4.072 | 0.577 | 0.680 |
| GVE-2606 | TM2 | 2003-2015 | 1.73 | | | | 0.237 | 0.030 | 0.045 | 5.480 | 0.094 | 0.728 | | |
| LUZ-2608 | TM1 | 2004-2019 | 0.79 | 0.21 | | | 3.595 | 0.467 | 0.843 | | 1.738 | 0.604 | | |
| EIN-2609 | TM1 | 2006-2017 | 1.16 | 2.28 | | | 1.126 | 0.452 | 0.426 | 0.571 | 6.228 | 4.985 | 0.579 | 0.895 |
| OTL-2612 | TM1 | 2004-2018 | 1.00 | 2.90 | | | -1.451 | 0.342 | 0.375 | | 3.369 | 2.738 | 0.613 | 0.448 |
| BAS-2613 | TM1 | 1995-2018 | 0.84 | | | | 0.297 | 0.036 | 0.054 | 12.982 | 0.114 | 0.611 | | |
| SMM-2617 | TM1 | 2003-2018 | 0.77 | 2.47 | | | 3.888 | 0.424 | 1.166 | -0.551 | | | | |
| ULR-2623 | TM1 | 2003-2019 | 0.70 | | | | 14.085 | 1.089 | 5.000 | | | | | |
| LUZ-2634 | TM1 | 1990-2016 | 0.75 | 15.30 | 1980-1989 | 0.74 | 1.187 | 0.789 | 0.852 | 0.141 | 5.369 | 3.959 | 0.590 | 0.739 |
| EIN-2635 | TM1 | 2003-2019 | 1.04 | 0.39 | | | 4.209 | 0.653 | 1.270 | | 3.516 | 0.593 | | |





**Table B3.** Spring (March to May) significant (p < 0.05) warming trends (°C decade$^{-1}$) for river measurements and best performing air2stream and air2water models with 30 years (1990-2019) of available data.

| Station | Thermal regime | air2stream | | | air2water | | |
|---|---|---|---|---|---|---|---|
| | | Measurements | Model | Bias | Measurements | Model | Bias |
| 2009 | Regulated | 0.17 | 0.08 | 0.09 | | | |
| 2016 | Downstream lake | | | | 0.20 | 0.21 | -0.01 |
| 2018 | Downstream lake | | | | 0.25 | 0.16 | 0.10 |
| 2044 | Swiss Plateau | 0.31 | 0.21 | 0.10 | | | |
| 2104 | Downstream lake | 0.23 | 0.11 | 0.12 | 0.23 | 0.26 | -0.03 |
| 2109 | Alpine | 0.23 | 0.09 | 0.14 | | | |
| 2113 | Downstream lake | | | | 0.23 | 0.17 | 0.06 |
| 2243 | Downstream lake | | | | 0.16 | 0.20 | -0.03 |
| 2372 | Regulated | 0.20 | 0.08 | 0.13 | | | |
| 2392 | Downstream lake | 0.18 | 0.21 | -0.04 | 0.18 | 0.20 | -0.03 |
| 2415 | Swiss Plateau | 0.20 | 0.17 | 0.03 | | | |
| 2473 | Regulated | 0.20 | 0.12 | 0.08 | | | |
| | | | Mean | | | Mean | |
| | All stations | 0.22 | 0.13 | 0.08 | 0.21 | 0.20 | 0.01 |
| | Downstream lake | 0.21 | 0.16 | 0.04 | 0.21 | 0.20 | 0.01 |
| | Regulated | 0.19 | 0.09 | 0.10 | | | |
| | Swiss Plateau | 0.25 | 0.19 | 0.07 | | | |
| | Alpine | 0.23 | 0.09 | 0.14 | | | |

**Table B4.** Summer (June to August) significant (p < 0.05) warming trends (°C decade$^{-1}$) for river measurements and best performing air2stream and air2water models with 30 years (1990-2019) of available data.

| Station | Thermal regime | air2stream | | | air2water | | |
|---|---|---|---|---|---|---|---|
| | | Measurements | Model | Difference | Measurements | Model | Difference |
| 2009 | Regulated | 0.14 | 0.05 | 0.09 | | | |
| 2016 | Downstream lake | 0.47 | 0.24 | 0.23 | 0.47 | 0.42 | 0.05 |
| 2018 | Downstream lake | 0.42 | 0.22 | 0.20 | 0.42 | 0.27 | 0.15 |
| 2019 | Regulated | 0.65 | 0.09 | 0.57 | | | |
| 2029 | Downstream lake | 0.40 | 0.31 | 0.08 | 0.40 | 0.29 | 0.11 |
| 2034 | Swiss Plateau | 0.54 | 0.49 | 0.05 | | | |
| 2044 | Swiss Plateau | 0.59 | 0.39 | 0.20 | | | |
| 2056 | Regulated | 0.30 | 0.09 | 0.21 | | | |
| 2070 | Swiss Plateau | 0.55 | 0.13 | 0.42 | | | |
| 2084 | Regulated | 0.14 | 0.09 | 0.05 | | | |
| 2104 | Downstream lake | 0.56 | 0.16 | 0.40 | 0.56 | 0.45 | 0.11 |
| 2106 | Swiss Plateau | 0.29 | 0.29 | 0.00 | | | |
| 2109 | Alpine | 0.66 | 0.09 | 0.57 | | | |
| 2113 | Downstream lake | 0.63 | 0.22 | 0.40 | 0.63 | 0.30 | 0.32 |
| 2135 | Downstream lake | 0.44 | 0.13 | 0.31 | 0.44 | 0.17 | 0.27 |
| 2152 | Downstream lake | 0.40 | 0.18 | 0.22 | 0.40 | 0.25 | 0.15 |
| 2176 | Swiss Plateau | 0.43 | 0.28 | 0.15 | | | |
| 2243 | Downstream lake | 0.47 | 0.24 | 0.23 | 0.47 | 0.37 | 0.10 |
| 2269 | Alpine | 0.34 | 0.03 | 0.31 | | | |
| 2372 | Regulated | 0.33 | 0.11 | 0.22 | | | |
| 2392 | Downstream lake | 0.58 | 0.49 | 0.09 | 0.58 | 0.44 | 0.14 |
| 2415 | Swiss Plateau | 0.47 | 0.23 | 0.24 | | | |
| 2473 | Regulated | 0.38 | 0.13 | 0.25 | | | |
| 2481 | Regulated | 0.24 | 0.08 | 0.16 | | | |
| 2500 | Swiss Plateau | 0.09 | 0.15 | -0.06 | | | |
| | | | Mean | | | Mean | |
| | All stations | 0.42 | 0.20 | 0.22 | 0.48 | 0.33 | 0.16 |
| | Downstream lake | 0.48 | 0.24 | 0.24 | 0.48 | 0.33 | 0.16 |
| | Regulated | 0.31 | 0.09 | 0.22 | | | |
| | Swiss Plateau | 0.42 | 0.28 | 0.14 | | | |
| | Alpine | 0.50 | 0.06 | 0.44 | | | |



**Table B5.** Autumn (September to November) significant (p < 0.05) warming trends (°C decade$^{-1}$) for river measurements and best performing air2stream and air2water models with 30 years (1990-2019) of available data.

| Station | Thermal regime | air2stream Measurements | Model | Difference | air2water Measurements | Model | Difference |
|---|---|---|---|---|---|---|---|
| 2009 | Regulated | 0.26 | 0.16 | 0.10 | | | |
| 2016 | Downstream lake | 0.45 | 0.23 | 0.23 | 0.45 | 0.29 | 0.16 |
| 2018 | Downstream lake | 0.47 | 0.19 | 0.28 | 0.47 | 0.19 | 0.28 |
| 2019 | Regulated | 0.40 | 0.05 | 0.35 | | | |
| 2029 | Downstream lake | 0.42 | 0.26 | 0.15 | 0.42 | 0.17 | 0.24 |
| 2034 | Swiss Plateau | 0.34 | 0.39 | -0.05 | | | |
| 2044 | Swiss Plateau | 0.50 | 0.28 | 0.22 | | | |
| 2056 | Regulated | 0.32 | 0.11 | 0.21 | | | |
| 2070 | Swiss Plateau | 0.34 | 0.14 | 0.20 | | | |
| 2104 | Downstream lake | 0.37 | 0.13 | 0.24 | 0.37 | 0.23 | 0.14 |
| 2106 | Swiss Plateau | 0.17 | 0.30 | -0.12 | | | |
| 2109 | Alpine | 0.44 | 0.13 | 0.31 | | | |
| 2113 | Downstream lake | 0.50 | 0.16 | 0.34 | 0.50 | 0.22 | 0.28 |
| 2152 | Downstream lake | | | | 0.45 | 0.17 | 0.28 |
| 2176 | Swiss Plateau | 0.31 | 0.31 | 0.00 | | | |
| 2243 | Downstream lake | | | | 0.45 | 0.28 | 0.17 |
| 2269 | Alpine | 0.15 | 0.07 | 0.08 | | | |
| 2372 | Regulated | 0.33 | 0.11 | 0.22 | | | |
| 2392 | Downstream lake | 0.54 | 0.37 | 0.17 | 0.54 | 0.31 | 0.24 |
| 2415 | Swiss Plateau | 0.31 | 0.19 | 0.12 | | | |
| 2473 | Regulated | 0.31 | 0.15 | 0.16 | | | |
| 2481 | Regulated | 0.25 | 0.08 | 0.18 | | | |
| | | | Mean | | | Mean | |
| | All stations | 0.36 | 0.19 | 0.17 | 0.46 | 0.23 | 0.22 |
| | Downstream lake | 0.46 | 0.22 | 0.24 | 0.46 | 0.23 | 0.22 |
| | Regulated | 0.31 | 0.11 | 0.20 | | | |
| | Swiss Plateau | 0.33 | 0.27 | 0.06 | | | |
| | Alpine | 0.29 | 0.10 | 0.19 | | | |

**Table B6.** Winter (December to February) significant (p < 0.05) warming trends (°C decade$^{-1}$) for river measurements and best performing air2stream and air2water models with 30 years (1990-2019) of available data.

| Station | Thermal regime | air2stream Measurements | Model | Difference | air2water Measurements | Model | Difference |
|---|---|---|---|---|---|---|---|
| 2009 | Regulated | 0.09 | 0.07 | 0.01 | | | |
| 2016 | Downstream lake | 0.27 | 0.13 | 0.14 | 0.27 | 0.23 | 0.04 |
| 2018 | Downstream lake | 0.29 | 0.11 | 0.19 | 0.29 | 0.14 | 0.15 |
| 2019 | Regulated | 0.08 | -0.03 | 0.12 | | | |
| 2029 | Downstream lake | 0.18 | 0.20 | -0.03 | 0.18 | 0.15 | 0.03 |
| 2034 | Swiss Plateau | 0.10 | 0.14 | -0.05 | | | |
| 2044 | Swiss Plateau | 0.33 | 0.14 | 0.19 | | | |
| 2084 | Regulated | 0.18 | 0.11 | 0.06 | | | |
| 2104 | Downstream lake | 0.19 | 0.11 | 0.07 | 0.19 | 0.24 | -0.06 |
| 2106 | Swiss Plateau | 0.09 | 0.13 | -0.05 | | | |
| 2109 | Alpine | 0.17 | 0.08 | 0.09 | | | |
| 2113 | Downstream lake | 0.17 | 0.12 | 0.05 | 0.17 | 0.16 | 0.01 |
| 2135 | Downstream lake | | | | 0.15 | 0.08 | 0.07 |
| 2152 | Downstream lake | 0.21 | 0.10 | 0.11 | 0.21 | 0.16 | 0.05 |
| 2243 | Downstream lake | 0.15 | 0.14 | 0.01 | 0.15 | 0.24 | -0.09 |
| 2372 | Regulated | 0.19 | 0.08 | 0.12 | | | |
| 2392 | Downstream lake | 0.23 | 0.29 | -0.06 | 0.23 | 0.25 | -0.02 |
| 2415 | Swiss Plateau | 0.09 | 0.12 | -0.03 | | | |
| 2473 | Regulated | 0.11 | 0.14 | -0.03 | | | |
| | | | Mean | | | Mean | |
| | All stations | 0.17 | 0.12 | 0.05 | 0.20 | 0.18 | 0.02 |
| | Downstream lake | 0.21 | 0.15 | 0.06 | 0.20 | 0.18 | 0.02 |
| | Regulated | 0.13 | 0.07 | 0.06 | | | |
| | Swiss Plateau | 0.15 | 0.13 | 0.02 | | | |
| | Alpine | 0.17 | 0.08 | 0.09 | | | |



**Table B7.** The mean difference between significant (p < 0.05) observed water temperature trends versus modeled trends (°C decade$^{-1}$) for air2stream an air2water at 25 stations. Differences have been averaged over available simulation and river stations from 1990 to 2019. Results are ordered according to the use of data from climate models or real measurements as atmospheric forcing for the water temperature models. Note that negative values indicate a larger mean modeled water temperature trend compared to the observed trend.

### All rivers

| | RCP8.5 | | RCP4.5 | | RCP2.6 | | Real measurements |
|---|---|---|---|---|---|---|---|
| | Corrected | No correction | Corrected | No correction | Corrected | No correction | No correction |
| All Year | -0.004 | 0.097 | 0.026 | 0.113 | 0.049 | 0.147 | 0.123 |
| March to May | -0.030 | 0.016 | 0.004 | 0.054 | 0.000 | 0.048 | 0.058 |
| June to August | 0.081 | 0.254 | 0.089 | 0.262 | 0.059 | 0.233 | 0.200 |
| September to November | -0.015 | 0.139 | -0.003 | 0.109 | 0.041 | 0.181 | 0.173 |
| December to February | -0.092 | -0.069 | -0.066 | -0.016 | -0.011 | 0.026 | 0.037 |

### Alpine

| | RCP8.5 | | RCP4.5 | | RCP2.6 | | Real measurements |
|---|---|---|---|---|---|---|---|
| | Corrected | No correction | Corrected | No correction | Corrected | No correction | No correction |
| All Year | -0.047 | 0.153 | -0.033 | 0.162 | -0.022 | 0.172 | 0.172 |
| March to May | -0.058 | 0.067 | -0.031 | 0.076 | -0.016 | 0.102 | 0.143 |
| June to August | 0.043 | 0.452 | 0.045 | 0.453 | 0.040 | 0.451 | 0.437 |
| September to November | 0.032 | 0.195 | 0.038 | 0.153 | 0.054 | 0.300 | 0.195 |
| December to February | -0.215 | -0.144 | -0.198 | -0.113 | -0.182 | -0.090 | 0.086 |

### Downstream Lake

| | RCP8.5 | | RCP4.5 | | RCP2.6 | | Real measurements |
|---|---|---|---|---|---|---|---|
| | Corrected | No correction | Corrected | No correction | Corrected | No correction | No correction |
| All Year | 0.003 | 0.106 | 0.056 | 0.132 | 0.081 | 0.164 | 0.125 |
| March to May | -0.059 | -0.049 | -0.022 | 0.008 | -0.028 | -0.012 | 0.014 |
| June to August | 0.124 | 0.267 | 0.131 | 0.272 | 0.117 | 0.272 | 0.175 |
| September to November | 0.000 | 0.192 | 0.031 | 0.177 | 0.110 | 0.248 | 0.232 |
| December to February | -0.100 | -0.083 | -0.066 | -0.032 | -0.001 | 0.022 | 0.032 |

### Regulated

| | RCP8.5 | | RCP4.5 | | RCP2.6 | | Real measurements |
|---|---|---|---|---|---|---|---|
| | Corrected | No correction | Corrected | No correction | Corrected | No correction | No correction |
| All Year | -0.030 | 0.096 | -0.004 | 0.110 | 0.003 | 0.136 | 0.136 |
| March to May | -0.007 | 0.065 | 0.017 | 0.082 | 0.027 | 0.095 | 0.098 |
| June to August | 0.003 | 0.198 | 0.030 | 0.220 | -0.001 | 0.195 | 0.220 |
| September to November | -0.047 | 0.150 | -0.020 | 0.114 | 0.005 | 0.127 | 0.201 |
| December to February | -0.054 | -0.020 | -0.069 | 0.019 | -0.017 | 0.049 | 0.056 |

### Swiss Plateau

| | RCP8.5 | | RCP4.5 | | RCP2.6 | | Real measurements |
|---|---|---|---|---|---|---|---|
| | Corrected | No correction | Corrected | No correction | Corrected | No correction | No correction |
| All Year | 0.026 | 0.071 | 0.035 | 0.077 | 0.074 | 0.129 | 0.093 |
| March to May | 0.010 | 0.046 | 0.051 | 0.090 | 0.021 | 0.069 | 0.066 |
| June to August | 0.114 | 0.237 | 0.107 | 0.236 | 0.051 | 0.158 | 0.143 |
| September to November | -0.015 | 0.045 | -0.041 | 0.003 | -0.014 | 0.161 | 0.060 |
| December to February | -0.078 | -0.072 | -0.026 | 0.001 | 0.031 | 0.046 | 0.015 |





**Table B8.** Mean temperature change from the reference period (1990 to 2019) to the near (2030 to 2059) and far future (2070 to 2099). Stations 2414 and 2462 are not shown since the flow model $M_4$ lacked 30 years of continuous data.

| Station | Near (Δ°C) | | | Far (Δ°C) | | |
|---|---|---|---|---|---|---|
| | RCP2.6 | RCP4.5 | RCP8.5 | RCP2.6 | RCP4.5 | RCP8.5 |
| **Alpine** | | | | | | |
| 2033 | 0.89 | 1.08 | 1.41 | 1.12 | 1.70 | 3.49 |
| 2109 | 0.95 | 1.08 | 1.40 | 1.30 | 1.82 | 3.55 |
| 2150 | 0.97 | 1.11 | 1.48 | 1.17 | 1.71 | 3.70 |
| 2161 | 0.70 | 0.89 | 1.09 | 0.91 | 1.48 | 2.61 |
| 2232 | 0.98 | 1.17 | 1.50 | 1.21 | 1.94 | 3.70 |
| 2256 | 0.77 | 1.04 | 1.34 | 0.89 | 1.78 | 3.49 |
| 2265 | 1.22 | 1.42 | 1.81 | 1.45 | 2.19 | 4.32 |
| 2269 | 0.77 | 1.03 | 1.24 | 0.97 | 1.71 | 2.96 |
| 2276 | 0.90 | 0.89 | 1.25 | 1.20 | 1.41 | 3.00 |
| 2327 | 0.79 | 1.08 | 1.34 | 0.96 | 1.73 | 3.19 |
| 2347 | 0.92 | 1.15 | 1.52 | 0.97 | 1.88 | 3.72 |
| 2351 | 1.02 | 1.12 | 1.46 | 1.43 | 2.04 | 4.05 |
| 2366 | 1.19 | 1.42 | 1.74 | 1.44 | 2.30 | 4.11 |
| 2617 | 1.09 | 1.32 | 1.67 | 1.26 | 2.10 | 3.92 |
| 2623 | 0.82 | 1.10 | 1.38 | 0.95 | 1.83 | 3.34 |
| Mean | 0.93 | 1.13 | 1.44 | 1.15 | 1.84 | 3.54 |
| **Downstream Lake** | | | | | | |
| 2016 | 0.79 | 1.03 | 1.32 | 0.86 | 1.64 | 3.37 |
| 2018 | 0.72 | 0.89 | 1.20 | 0.74 | 1.44 | 3.00 |
| 2029 | 1.13 | 1.10 | 1.48 | 1.33 | 1.74 | 3.76 |
| 2030 | 0.87 | 0.75 | 1.03 | 1.06 | 1.17 | 2.58 |
| 2085 | 1.01 | 0.94 | 1.26 | 1.30 | 1.45 | 3.26 |
| 2091 | 0.86 | 0.98 | 1.33 | 0.88 | 1.56 | 3.38 |
| 2104 | 1.09 | 1.18 | 1.61 | 1.17 | 1.85 | 4.08 |
| 2113 | 0.77 | 0.94 | 1.30 | 0.79 | 1.53 | 3.25 |
| 2130 | 0.88 | 0.92 | 1.28 | 1.00 | 1.49 | 3.18 |
| 2135 | 0.85 | 0.74 | 1.04 | 1.04 | 1.16 | 2.62 |
| 2143 | 0.97 | 1.20 | 1.57 | 0.99 | 1.94 | 3.95 |
| 2152 | 0.86 | 0.93 | 1.31 | 0.91 | 1.43 | 3.32 |
| 2167 | 1.00 | 1.16 | 1.51 | 0.99 | 1.78 | 3.75 |
| 2174 | 0.88 | 0.87 | 1.21 | 1.05 | 1.37 | 3.00 |
| 2243 | 0.84 | 1.04 | 1.36 | 0.85 | 1.68 | 3.43 |
| 2288 | 1.03 | 1.29 | 1.68 | 1.02 | 2.07 | 4.28 |
| 2392 | 0.97 | 1.19 | 1.61 | 0.96 | 1.91 | 4.03 |
| 2457 | 0.77 | 0.87 | 1.21 | 0.78 | 1.39 | 3.03 |
| 2606 | 1.09 | 1.11 | 1.48 | 1.29 | 1.72 | 3.78 |
| 2613 | 0.84 | 1.00 | 1.39 | 0.85 | 1.62 | 3.48 |
| Mean | 0.91 | 1.01 | 1.36 | 0.99 | 1.60 | 3.43 |
| **Regulated** | | | | | | |
| 2009 | 0.93 | 0.90 | 1.15 | 1.30 | 1.52 | 3.21 |
| 2019 | 0.69 | 0.70 | 1.00 | 0.88 | 1.09 | 2.43 |
| 2056 | 0.77 | 0.77 | 1.07 | 0.97 | 1.23 | 2.71 |
| 2068 | 0.84 | 1.01 | 1.33 | 0.89 | 1.59 | 3.20 |
| 2084 | 0.86 | 0.95 | 1.23 | 1.06 | 1.43 | 3.14 |
| 2372 | 0.71 | 0.71 | 0.99 | 0.93 | 1.08 | 2.45 |
| 2467 | 1.00 | 1.27 | 1.70 | 1.09 | 2.06 | 4.24 |
| 2473 | 0.80 | 0.90 | 1.15 | 0.92 | 1.31 | 2.92 |
| 2481 | 0.65 | 0.78 | 1.08 | 0.78 | 1.23 | 2.68 |
| Mean | 0.80 | 0.89 | 1.19 | 0.98 | 1.39 | 3.00 |
| **Swiss Plateau** | | | | | | |
| 2034 | 0.90 | 1.11 | 1.39 | 1.02 | 1.73 | 3.61 |
| 2044 | 0.75 | 1.06 | 1.29 | 0.83 | 1.64 | 3.35 |
| 2070 | 0.60 | 0.78 | 0.99 | 0.72 | 1.24 | 2.57 |
| 2106 | 0.66 | 0.88 | 1.09 | 0.72 | 1.37 | 2.84 |
| 2112 | 0.59 | 0.81 | 1.04 | 0.63 | 1.27 | 2.72 |
| 2126 | 0.52 | 0.71 | 0.89 | 0.62 | 1.11 | 2.27 |
| 2139 | 0.46 | 0.58 | 0.77 | 0.51 | 0.93 | 1.92 |
| 2159 | 0.62 | 0.86 | 1.08 | 0.70 | 1.35 | 2.82 |
| 2170 | 0.47 | 0.58 | 0.76 | 0.57 | 0.96 | 1.88 |
| 2176 | 0.77 | 1.07 | 1.33 | 0.85 | 1.70 | 3.47 |
| 2179 | 0.69 | 0.92 | 1.16 | 0.77 | 1.47 | 3.05 |
| 2181 | 0.78 | 1.09 | 1.39 | 0.84 | 1.68 | 3.61 |
| 2210 | 0.66 | 0.92 | 1.13 | 0.70 | 1.45 | 2.97 |
| 2282 | 0.56 | 0.75 | 0.98 | 0.61 | 1.24 | 2.47 |
| 2307 | 0.42 | 0.64 | 0.72 | 0.47 | 0.99 | 1.97 |
| 2308 | 0.72 | 1.00 | 1.30 | 0.79 | 1.59 | 3.38 |
| 2343 | 0.46 | 0.63 | 0.79 | 0.52 | 1.05 | 2.01 |
| 2369 | 0.75 | 0.95 | 1.20 | 0.86 | 1.54 | 3.12 |
| 2374 | 0.73 | 0.98 | 1.22 | 0.83 | 1.53 | 3.19 |
| 2386 | 0.64 | 0.87 | 1.08 | 0.72 | 1.35 | 2.78 |
| 2410 | 0.36 | 0.44 | 0.60 | 0.43 | 0.71 | 1.48 |
| 2415 | 0.57 | 0.73 | 0.94 | 0.64 | 1.20 | 2.37 |
| 2432 | 0.72 | 0.96 | 1.21 | 0.77 | 1.51 | 3.11 |
| 2433 | 0.61 | 0.81 | 1.05 | 0.70 | 1.32 | 2.64 |
| 2434 | 0.68 | 0.98 | 1.22 | 0.74 | 1.54 | 3.11 |
| 2485 | 0.50 | 0.68 | 0.87 | 0.56 | 1.14 | 2.18 |
| 2493 | 0.54 | 0.74 | 0.94 | 0.59 | 1.17 | 2.38 |
| 2500 | 0.52 | 0.66 | 0.83 | 0.64 | 1.05 | 2.15 |
| 2604 | 0.71 | 0.90 | 1.15 | 0.81 | 1.44 | 3.06 |
| 2608 | 0.64 | 0.83 | 1.11 | 0.71 | 1.36 | 2.80 |
| 2609 | 0.71 | 0.94 | 1.19 | 0.84 | 1.48 | 3.13 |
| 2612 | 0.69 | 0.90 | 1.15 | 0.67 | 1.45 | 2.99 |
| 2634 | 0.76 | 1.03 | 1.31 | 0.88 | 1.59 | 3.44 |
| 2635 | 0.63 | 0.76 | 1.01 | 0.72 | 1.25 | 2.57 |
| Mean | 0.63 | 0.84 | 1.06 | 0.71 | 1.33 | 2.75 |
| **Spring** | | | | | | |
| 2290 | 0.06 | 0.08 | 0.09 | 0.06 | 0.12 | 0.24 |
| 2499 | -0.01 | -0.01 | -0.02 | -0.01 | -0.02 | -0.05 |
| Mean | 0.02 | 0.03 | 0.04 | 0.03 | 0.05 | 0.10 |





**Table B9.** Change in hysteresis classes marked by yellow from the reference period (1990 to 2019) to the near (2030 to 2059)
and the far future (2070 to 2099) using climate scenario RCP4.5. Flow data from models $M_2$, $M_3$ and $M_4$. Stations with no flow
measurements for calibration, missing flow model output as forcing or where the use of the air2water model did not require
flow as input have been excluded. A change in class from the reference period to the near or far future period is highlighted in
*italic*.

| Station | Reference | | | Near | | | Far | | |
|---|---|---|---|---|---|---|---|---|---|
| | $M_1$ | $M_2$ | $M_3$ | $M_1$ | $M_2$ | $M_3$ | $M_1$ | $M_2$ | $M_3$ |
| **Downstream Lake** | | | | | | | | | |
| 2016 | 4 | | | 4 | | | 4 | | |
| 2085 | 4 | | | 4 | | | 4 | | |
| **Regulated** | | | | | | | | | |
| 2009 | 3 | | | 3 | | | *4* | | |
| 2056 | 3 | 3 | | 3 | 3 | | 3 | 3 | |
| 2084 | | 4 | | | 4 | | | 4 | |
| 2372 | 4 | 4 | | 4 | 4 | | 4 | 4 | |
| 2473 | 3 | | | 3 | | | *4* | | |
| 2481 | | 4 | 4 | | 4 | 4 | | 4 | 4 |
| **Swiss Plateau** | | | | | | | | | |
| 2034 | -1 | -1 | | -2 | -2 | | -2 | -2 | |
| 2044 | 4 | 4 | 4 | -2 | -2 | -2 | -2 | -2 | -2 |
| 2070 | 4 | 4 | | 4 | 4 | | 4 | 4 | |
| 2106 | -1 | -2 | | -2 | -2 | | -2 | -2 | |
| 2112 | | 4 | | | 4 | | | 4 | |
| 2126 | | -1 | | | -1 | | | -2 | |
| 2159 | | 3 | | | -2 | | | -2 | |
| 2176 | 4 | 4 | | *3* | 4 | | 4 | 4 | |
| 2179 | 4 | 4 | | 4 | 4 | | 4 | 4 | |
| 2181 | 4 | 4 | | *-1* | 4 | | *-1* | 4 | |
| 2210 | | -2 | | | -2 | | | -2 | |
| 2307 | -1 | -1 | | -1 | -2 | | -1 | -2 | |
| 2308 | | 4 | | | -2 | | | -2 | |
| 2343 | | -1 | | | -1 | | | -1 | |
| 2369 | | -1 | | | -2 | | | -2 | |
| 2374 | | 4 | | | -2 | | | -2 | |
| 2386 | | -2 | | | -2 | | | -2 | |
| 2415 | -2 | -2 | | -2 | -2 | | -2 | -2 | |
| 2432 | -1 | -1 | | -2 | -1 | | -1 | -2 | |
| 2434 | | -1 | | | -1 | | | -1 | |
| 2493 | | -1 | | | -1 | | | -1 | |
| 2500 | | -1 | | | -1 | | | -1 | |
| 2604 | | 4 | | | 4 | | | 4 | |
| 2609 | | 4 | | | 4 | | | 4 | |
| 2612 | | 3 | | | 3 | | | 3 | |
| 2634 | | 4 | 4 | | 4 | 4 | | 4 | 4 |
| **Alpine** | | | | | | | | | |
| 2033 | 3 | 3 | | 3 | 3 | | 3 | 3 | |
| 2109 | 3 | | 3 | 3 | | 3 | 3 | | 3 |
| 2150 | 4 | | | 4 | | | 4 | | |
| 2161 | 1 | | 1 | 1 | | 1 | *2* | | *2* |
| 2232 | | 4 | | | 4 | | | 4 | |
| 2256 | | 3 | | | 3 | | | 3 | |
| 2265 | 3 | | | 3 | | | 3 | | |
| 2269 | | | 4 | | | 4 | | | 4 |
| 2276 | | 4 | 4 | | 4 | 4 | | 4 | *3* |
| 2327 | | | 3 | | | 3 | | | 3 |
| 2351 | 3 | | | 3 | | | 3 | | |
| 2366 | | 3 | 3 | | 3 | 3 | | 3 | 3 |
| 2617 | | 3 | 3 | | 3 | 3 | | 3 | 3 |





**Table B10.** Change in hysteresis classes marked by yellow from the reference period (1990 to 2019) to the near (2030 to 2059)
and the far future (2070 to 2099) using climate scenario RCP2.6. Flow data from models $M_2$, $M_3$ and $M_4$. Stations with no flow
measurements for calibration, missing flow model output as forcing or where the use of the air2water model didn't require flow
as input have been excluded. A change in class from the reference period to the near or far future period is highlighted in *italic*.

| Station | Reference | | | Near | | | Far | | |
|---|---|---|---|---|---|---|---|---|---|
| | $M_1$ | $M_2$ | $M_3$ | $M_1$ | $M_2$ | $M_3$ | $M_1$ | $M_2$ | $M_3$ |
| **Downstream Lake** | | | | | | | | | |
| 2016 | 3 | | | *4* | | | *4* | | |
| 2085 | 4 | | | 4 | | | 4 | | |
| **Regulated** | | | | | | | | | |
| 2009 | 3 | | | 3 | | | *4* | | |
| 2056 | 3 | 3 | | *4* | *4* | | *4* | *4* | |
| 2084 | | 4 | | | 4 | | | 4 | |
| 2372 | 4 | 4 | | 4 | 4 | | 4 | 4 | |
| 2473 | 3 | | | *4* | | | *4* | | |
| 2481 | | 4 | 4 | | 4 | 4 | | 4 | 4 |
| **Swiss Plateau** | | | | | | | | | |
| 2034 | -1 | -1 | | -2 | -2 | | -2 | -2 | |
| 2044 | 4 | 4 | 4 | -2 | -2 | -2 | 4 | 4 | 4 |
| 2070 | 4 | 4 | | 4 | 4 | | 4 | 4 | |
| 2106 | -2 | -2 | | -2 | -2 | | -2 | -2 | |
| 2112 | | 4 | | | 4 | | | 4 | |
| 2126 | | -1 | | | -2 | | | -2 | |
| 2159 | | 4 | | | 4 | | | 4 | |
| 2176 | 4 | 3 | | 4 | *4* | | 4 | 3 | |
| 2179 | 4 | 4 | | 4 | 4 | | 4 | 4 | |
| 2181 | 4 | 4 | | 4 | 4 | | 4 | 4 | |
| 2210 | | -2 | | | -2 | | | -2 | |
| 2307 | -1 | -1 | | -1 | -2 | | -1 | -1 | |
| 2308 | | -2 | | | *3* | | | -2 | |
| 2343 | | -1 | | | -1 | | | -1 | |
| 2369 | | -1 | | | -1 | | | -1 | |
| 2374 | | 4 | | | -2 | | | -2 | |
| 2386 | | -2 | | | -2 | | | -2 | |
| 2415 | -2 | -2 | | -2 | -2 | | -2 | -2 | |
| 2432 | -1 | -1 | | -1 | -1 | | -1 | -1 | |
| 2434 | | -1 | | | -1 | | | -1 | |
| 2493 | | -1 | | | -1 | | | -1 | |
| 2500 | | -1 | | | -1 | | | -1 | |
| 2604 | | 4 | | | 4 | | | 4 | |
| 2609 | | 4 | | | 4 | | | 4 | |
| 2612 | | 3 | | | 3 | | | 3 | |
| 2634 | | 4 | 4 | | 4 | 4 | | 4 | 4 |
| **Alpine** | | | | | | | | | |
| 2033 | 3 | 3 | | *4* | *4* | | *4* | *4* | |
| 2109 | 3 | | 3 | *4* | | 3 | *4* | | 3 |
| 2150 | 4 | | | 4 | | | 4 | | |
| 2161 | 1 | | 1 | 1 | | *2* | 1 | | 1 |
| 2232 | | 4 | | | 4 | | | 4 | |
| 2256 | | 3 | | | 3 | | | 3 | |
| 2265 | 3 | | | 3 | | | 3 | | |
| 2269 | | | 4 | | | 4 | | | 4 |
| 2276 | | 4 | 4 | | 4 | 4 | | 4 | 4 |
| 2327 | | | 3 | | | 3 | | | 3 |
| 2351 | 3 | | | 3 | | | *4* | | |
| 2366 | | 3 | 3 | | 3 | 3 | | 3 | 3 |
| 2617 | | 3 | 3 | | 3 | 3 | | 3 | 3 |

1057

1058