# Peer review of "Multi-fidelity model assessment of climate change impacts on river water temperatures, thermal extremes and potential effects on cold water fish in Switzerland"

_EGUsphere, 2024_

## Author Comment (AC1)

We thank the reviewer for constructive comments and provide our answers hereunder.

**RC1**

RC1: 'Comment on egusphere-2024-3957', Anonymous Referee #1, 18 Mar 2025 reply

This is my review of "Multi-fidelity model assessment of climate change impacts on river water temperatures, thermal extremes and potential effects on cold water fish in Switzerland" by Love Raman Vinna et al., submitted to Hydrology and Earth System Sciences. In this paper, the authors combine an ensemble of climate models, hydrological models and two water temperature models with varying parameter settings per station to derive water temperature projections in Swiss rivers. In addition, they determined future changes in projected extremes, hysteresis effects and impacts on brown trout. I found the study interesting, with sound methodology. However, the manuscript would benefit from a clearer presentation to fully convey its strengths. I therefore suggest minor revisions, and I provide comments below to help improve these aspects.

Major comments:

While comprehensive and detailed, the data and methods section is somewhat difficult to follow due to its length and repetition. I recommend a thorough revision to improve its structure and conciseness, eliminating redundant wording. This will enhance readability and streamline the section. Below, I provide specific examples and suggestions to support this refinement.

Section 2 has been adjusted to make it clearer for the reader, see below for specific changes.

The structure has been improved by separating the entire text into more thematically related paragraphs.

The "Material & Methods"-section now is written more concise resulting in a text reduction of 3725 to 1529 characters (with spacings), following the suggestion of the reviewer one paragraph has been moved to the "Introduction"-section.

The "Data"-section now is written more concise by omitting redundant wording on used circulation and hydrological models.

The presentation of results could be refined to enhance clarity and readability. Figures 4, 6, and 7, along with Table 3, contain a wealth of information, but an additional or alternative figure presenting the data in a more aggregated way—such as by thermal regime—could help highlight key differences more effectively while still accounting for uncertainty. While the station-based results provide valuable detail, incorporating more synthesized figures or tables could make the main findings more accessible. Additionally, summarizing key insights more narratively, rather than listing numbers extensively, may improve the flow of the results section.

We prepared a figure that summarizes the entire range of individual evaluations for each regime. We do not consider this to be the right approach and would prefer to omit this type of presentation of the results.

Specific comments:

Title: the "cold water fish" might suggest a more elaborate analysis on general fish species when only brown trout is considered in the analysis. Suggestion to rephrase/remove.

Title changed to: Multi-fidelity model assessment of climate change impacts on river water temperatures, thermal extremes and potential effects on brown trout in Switzerland

L 16-18, abstract: Provide more detailed "Alpine thermal regime" and "Downstream lake regime".

Results have been added to the Abstract: Line 13 to 19 in revised manuscript now reads>

Results show that, until the end of the 21$^{st}$ century, average river water temperatures in Switzerland will likely increase by 3.1±0.7 °C (or 0.36±0.1 °C per decade) under RCP8.5, while under RCP2.6 the temperature increase may remain at 0.9±0.3 °C (0.12±0.1 °C per decade). Under RCP8.5, temperatures of rivers classified as being in the *Alpine* thermal regime will increase the most, that is, by 3.5±0.5 °C, followed by rivers of the *Downstream Lake* regime, which will increase 3.4±0.5 °C. Under RCP2.6 temperatures in the Alpine and Downstream lake regimes change most with +1.15 and +0.99±0.5 °C.

The introduction could benefit from a better gap description, and background building up to this gap description, for example highlighting the gaps of current water temperature projections for Switzerland available in literature. Also, since part of the novelty of the study lies in the modelling approach employed, this gap can also be made more apparent.

Added to introduction line 99: Compared to previous projections of climate warming in Swiss rivers (Michel et al., 2022), the simplified multi-fidelity modelling approach enables a wider investigation area (+90%) including 5 thermal regimes (previous 2) and 22 GCM-RCM chains (previously 7).

Figure 1: Good overview figure, although it would benefit from a distinction between data and data sources and operations on that data in the flow chart. This is an open suggestion

Additional detail has been added to the figure in the revised manuscript.

L73-92: this is a very general explanation about the choice of models to use, I would suggest to move it to the intro

We agree and have moved this to section 1 lines 60 to 79.

L126-138: can you give a little more extensive description of the CH2018 climate data? E.g. introduce that they are derived from the EURO-CORDEX regional climate modelling ensemble, with the number of RCMs driven by GCMs and future scenarios, as well as the horizontal resolution. These names return then in Table 1, allowing the reader to understand where they come from.

Added to line 145: Here we use CH2018 climate simulations based on the EURO-CORDEX regional climate modelling ensemble, for which near-surface air temperatures have been downscaled to local conditions with quantile mapping (CH2018, 2018). CH2018 comprises

simulations of 9 GCM coupled to 8 RCM runs for a total of 22 GCM-RCM model chains with 0.11° and 0.44° resolution under 3 climate change scenarios (RCP2.6, 4.5, and 8.5).

L130- ...It would help to also introduce the Hydro-CH2018 in a little bit more detail afterwards, with a subclause per hydrological model on its characteristics (semi-distributed, empirical etc). A suggestion is to structure the description of the different data sources with different subtitles

Section 2.1 has been updated to give more information of CH2018 and Hydro-CH2018 in line with additional reviewer comments hereunder. Line 145 to 164 now reads.

Here we use CH2018 climate simulations based on the EURO-CORDEX regional climate modeling ensemble. In CH2018 near-surface air temperatures was downscaled by applying a statistical bias-correction and downscaling method (Quantile Mapping, a purely statistical and data-driven method) to the original output of all EURO-CORDEX climate model simulations, as observational reference station observations and observation-based gridded analyses were used (CH2018, 2018, Chapter 5). These data are available as both gridded and local station products (CH2018 Project Team, 2018). Following CH2018, the Hydro-CH2018 project analyzed the effects of climate change on Swiss water bodies (FOEN, 2021). The gridded climate product from CH2018 was used to construct projections of future river discharge for 4 hydrological models used in Hydro-CH2018. The location where output from these 4 models was used in this study is shown in Figure 2a including: ($M_1$) PREVAH-WSL a conceptual process-based model (Brunner, et al., 2019a; Brunner, et al., 2019b) and ($M_2$) PREVAH-UniBE (Muelchi et al., 2021), ($M_3$) HBV Light-UniZH a bucket-type hydrological model (Freudiger et al., 2021), and ($M_4$) AlpineFlow-EPFL the snowmelt and runoff model Alpine3D coupled to the semi-distributed hydrological model StreamFlow (Michel et al., 2022). The Hydro-CH2018 project produced projections for 61 out of the 82 FOEN river monitoring stations under 22 GCM-RCM model chains (9 GCM coupled to 8 RCM runs) with 0.11° and 0.44° resolution and 3 climate change scenarios (RCP2.6, 4.5, and 8.5). The available projections, the employed circulation and hydrological models, and the considered climate change scenarios for all the different stations that were considered in this study are summarized in Table 1.

L151: I might have missed it, but how many monitoring stations are eventually used in the study? If these are the stations on Fig. 1 panel a, provide a reference to that fig (also for its other panels).

We used in total 82 stations (Figure 2 a and b). This is described in section 2, which has been revised for clarity. Line 113 to 116 now reads.

All available model configurations (i.e., 3,4,5,6,7 and 8 different parameter combinations and implementations) were evaluated for their applicability to different thermal river regimes (Appendix A) and allowed for developing optimal site-specific models for all the 82 thermal river monitoring stations of the Swiss Federal Office of the Environment (FOEN).

L167: DIS criterion: what is the threshold used to assume horizontal distance is "minimal"? & L168: how is the representativeness of the meteorological stations to upstream drainage area assessed?

Line 191 to 201 in Section 2.2 now reads:

Meteorological stations were subsequently paired with hydrological stations such that (a) the horizontal distance between river and meteorological stations was as small as possible i.e. nearest to nearest (criterion *DIS*), (b) the meteorological station was representative for the conditions in the upstream drainage area composing a meteorological station being located in the same valley and upstream (criterion *DRA*), and (c) the elevation difference did not exceed a reasonable threshold of 200 m (criterion *ELE*). Where possible, all three criteria were met, that is the closest station passed both *ELE* and *DRA* and are noted as *DIS* in Table 2. If the closest station were deemed not to be representative (e.g. in a neighboring valley or downstream) the *DIS* criteria where failed, such a station are noted as *DRA* in Table 2. If a station failed both *DIS* and *DRA* but passed *ELE* it is noted as *ELE* in Table 2. Station details and pairings are summarized in Table 2.

Table 2: I would suggest to move this table to the appendix, as it is very extensive and does not add much to the results.

Table 2 has been moved to appendix

L179: "were already statistically downscaled", please add details on how this is done. This could be part of the paragraph where the CH2018 scenarios are explained in more detail. Also, how big is the bias, and how much would it impact the results?

See adjusted text above regarding the paragraph where the CH2018 scenarios are explained. The bias for air temperature was small from station to station and while for the projections of future flow it was substantial compared to measurements in the reference period. So large in fact that we needed to the bias correction in section 2.3 in order to apply air2stram and air2water.

L222-228, suggestion to move this paragraph to after the model description.

Paragraph moved to end of section 2.5.Line 295 to 301.

L248-250: The fact that for each river monitoring station, the best water temperature model is employed is a key strength of the study, and should, to my opinion, be more pronounced throughout the study (eg intro describing the gap on this, abstract and earlier in the methods where the multi-fidelity is mentioned first).

Added to section 1 line 102 to 107

By grouping catchments together via statistical pattern recognition, we were able to classify rivers (including spring-fed rivers) into 5 different thermal regimes, improving model results by allowing for optimal model selection at each station and enabling regime-specific analyses. The effect on warming by changing river discharge was investigate through a hysteresis analysis. Additionally, we introduce the *extreme event severity* index as an analytic tool to evaluate the change in thermal extreme amplitude.

L287-320: there is some repetition in this section, and parts are more difficult to follow, please consider condensing it retaining the same information

We have rewritten this part. Line 313 to 352 now reads

Both *air2stream* and *air2water* underestimate the annual temperature trend during the reference period on average by 0.14 and 0.11 °C per decade, respectively. For *air2stream*, the annual trend bias is smallest for the *Swiss Plateau* thermal regime (0.09 °C per decade) and largest in the *Alpine* thermal regime (0.17 °C per decade). Seasonally, the trend bias is largest from June to August and September to November, whereas, especially for *air2water*, the bias is small from December to February and March to May.

The divergence of both *air2stream* and *air2water* models from observed trends warrant a post simulation bias correction of simulated trends. The bias is river station dependent, making an individual correction at each station preferable (Tables B3 to B6 in Appendix B). However, only about 30% of the river stations investigated have long enough data sets (30 years) for individual correction. Therefore, we tied the seasonal trend bias correction to the thermal regime, thereby keeping the correction linked to local conditions. Note that no river station of the *Spring* thermal regime had enough data to allow for the trend bias correction. *Spring* river stations were therefore not trend bias corrected. As the trend bias correction is acting on climate simulations of river temperature stretching from 1990 to 2099, the bias correction had to be scaled towards how air temperature trends shift in the climate models. The scaling was designed such that it did not affect the bias correction during the reference period (1990 to 2019), while adjusting the correction towards how the air temperature trend (*TTair*) changes in the near- (2030 to 2059) and far-future (2070 to 2099). For this purpose, an adjustment factor *Fs* (-) was constructed from the mean climate models air temperature trends for each climate scenario. *Fs* is thus specific for each climate scenario, river station and season.

$$Fs_{i,s} = \frac{TTair_{i,s}}{TTair_{ref,s}} \tag{2}$$

Here $TTair_{i,s}$ is the mean of the air temperature trends from the climate models, which is changing for each season and with the reference, near- and far-future periods, $TTair_{ref,s}$ is the mean of the seasonal air temperature trend during the reference period, *i* is the number of days, and *s* denotes the season. The temporal gaps between 1990 to 2019, 2030 to 2059 and 2070 to 2099, during which the air temperature trends were calculated, were linearly filled with shape-preserving piecewise cubic interpolation resulting in a continuous *factor* $Fs_{i,s}$ from 1990 to 2099. $Fs_{i,s}$ varied from -2 to +3 depending on the season and climate scenario and was applied for simulations using discharge input from models $M_1$ to $M_3$, while for simulations using $M_4$, $Fs_{i,s}$ was set to 1 from 1990 to 2099 due to too short simulation time frames in $M_4$ (only one decade). With $Fs_{i,s}$, the seasonal and thermal regime dependent water temperature bias $Tb_{i,s}$ (regime dependent mean from Table C3 to C6 in Appendix C) is turned into the thermal regime and climate scenario dependent seasonal bias correction $Bc_s$ (°C day$^{-1}$)

$$Bc_s = \sum_{i=1}^{i=n} Fs_{i,s} * Tb_{i,s} \tag{3}$$

where *n* is the number of days since 1$^{st}$ of January 1990. Before adjusting the water temperature model output from 1990 to 2099, the seasonal $Bc_s$ was combined into a continuous dataset *Bc*. To avoid a sharp shift in *Bc* between each season, a 3- to 5-day gap in between each season was smoothed with shape-preserving interpolation (Piecewise cubic Hermite interpolation, PCHIP; Matlab R2022a).

L332: could you give a more explicit explanation of a hysteresis example here, relevant for Swiss rivers?

Yes, paragraph has been updated on line 357 to 365

Hysteresis, wherein a dependent variable (water temperature or suspended sediments) can exhibit multiple values in response to a single value from the independent variable (discharge), is a common phenomenon in hydrology (Gharari & Razavi, 2018). Sediment transport hysteresis can be caused in rivers by emptying and refilling of sediment layers on the river bed (Tananaev, 2012) and through erosion on land as shown in the Alps with the contributing location (river bed or eroded area) determining the hysteresis loop shape and rotation direction (Misset et al., 2019). Stream temperature can also show hysteresis effects, example being a lag in the response to air temperature caused by ice-melt or reservoir release (Van Vliet et al., 2011; Webb & Nobilis, 1994).

L357-361: this is a good example of lines that could be shortened.

Now on line 390 to 393

Extreme conditions depend on what is considered to be extreme in relation to normal conditions (Stephenson, 2008). Here, water temperatures are considered to be extremely high if they exceed the $90^{th}$ percentile during the 30-year reference, near- and far-future periods (IPCC, 2014).

L368-372: if an extreme is defined by the deviation of the 90th percentile compared to the median of a certain 30 year period, why does this period need to be detrended? i.e., there is no certain time (beginning or end of period) where the analysis is carried out? Or am I missing something?

Within a 30-year period (1990 to 2019, 2030 to 2059, 2070 to 2099) small trends from the underlying climate may exist. If such a trend is observable and positive, it means that at the end of the 30-year period it will be easier for a temperature event to be above the $90^{th}$ percentile of the 30-year period compared to during the beginning of the period. These high temperature events could potentially incorrectly be classified as an extreme. By detrending within these 30 years, each extreme candidate will be considered irrespectively of whether it is in the beginning or at the end of the 30 years period.

L396-400: If this info would be presented in a small table, it would be easier to grasp Section 3.2 on hysteresis analysis. For a non-expert in hysteresis, I found the results difficult to interpret. To my opinion, it would be beneficial to provide some guiding sentences on how these results could be interpreted.

Sections 3.2 have been reworked to make it easier for the reader. Line 454 to 486

The hysteresis class could be determined for each station with future and present river discharge (47 out of 82 stations). For all stations, climate scenarios, and climate models, the index found solutions in hysteresis intervals ranging from 164 to 328 days.

During the reference period the dominant hysteresis class was IV (45.6%) followed by III (25.0%), -I (14.7%), -II (11.8%) and I (2.9%) while no stations belonged to class II. For the reference period the classes remained independent in relation to the climate scenario (RCP8.5, 4.5, 2.6) or hydrological model (M1, M2, M3) used, while in the near- and far-future differences start to show. For RCP8.5 in the far-future period the dominant class was -I (48.5%) followed by class IV (33.8%), III (13.2%) and -II (4.4%).

For the RCP8.5 scenario classes are shown for the reference, near- and far-future periods in Table 3 (hysteresis classes for RCP4.5 are shown in Table B9, and for RCP2.6 in Table B10, both in Appendix B). Under RCP8.5, the number of stations which changed hysteresis classes between the reference and the near-future was 23%, increasing to 51% until the far-future. Correspondingly, under RCP4.5, 23% had changed hysteresis classes when reaching the near-future, while 38% of the stations changed classes until the far-future. Under RCP2.6, 28% of stations had changed classes until the near-future, but once reaching the far-future, some stations changed back again and the fraction of stations that were in a different hysteresis class compared to the reference period was reduced to 21%.

Considering only the far-future period (2070 to 2099) , stations belonging to the Swiss Plateau thermal regime showed the largest change in hysteresis loop classes, with 58% changing under RCP8.5, 42% under RCP4.5 and 12% under RCP2.6. Considering again only the far-future, stations belonging to the Regulated thermal regime exhibited hysteresis loop class changes of 50% under RCP8.5, 33% under RCP4.5 and 50% under RCP2.6. Least prone to hysteresis class changes in the far-future were stations of the Alpine thermal regime (38% under RCP8.5 and RCP4.5, 23% under RCP2.6). Out of the 20 Downstream Lake thermal regime stations only 2 stations were investigated with discharge (i.e. model with air2stream instead of air2water). From these 2 stations, 1 changed hysteresis class with RCP8.5 by the far-future, 1 with RCP2.6 but none with RCP4.5. As can be seen from 4 representative stations for the Swiss Plateau, Regulated, Alpine, and Downstream Lake illustrated in Figure 5, a change in hysteresis class is usually associated with a counterclockwise rotation and stretching of the loop from, for example from a lower to a higher class (III to IV). Such a rotation and stretching appears as a result of increased warming in summer combined with a decrease in summer discharge, while warming in winter is smaller than in summer and discharge is increasing.

Section 3.3 and figure 6. It should be more clear from the start of the paragraph that the "extreme event severity index" is used, so the values do not represent absolute extremes, but deviations from the "normal" in the respective period. L477-479 indicate this, but it should be more up front in the paragraph and figure to avoid confusion for the reader.

Section 3.3 has been updated following the reviewer's comment.   Line 495 to 523

The analysis is focused on temperature extremes in the summer months (June to August), during which the severity of extremes varies in between climate scenarios and is different on individual station basis and on a thermal regime basis (Figure 6). Note that the use of extreme event severity as an index should be viewed as the minimum temperature increase of extreme events in the future while it denotes the increase of the 90[th] percentile. From the reference (1990 to 2019) to the far-future (2070 to 2099) period the extreme event severity for scenario RCP2.6 increased on average by +0.20 °C (Figure 6a), by +0.38 °C for RCP4.5 (Figure 6 b) and by +0.61 °C for RCP8.5 (Figure 6 c).

Looking at extreme events at the level of thermal regimes, during the reference period (1990 to 2019), the most sever extreme temperatures occurred at stations in the *Swiss Plateau* and *Downstream Lake* thermal regimes. *Swiss Plateau* thermal regime (mean extreme event severity +2.8 °C) *Downstream Lake* (+2.2 °C), *Regulated* (+1.3 °C), *Alpine* (+1.1 °C) and *Spring* thermal regimes (+0.12 °C).

For all climate scenarios and all thermal regimes, the severity of extreme events increased throughout the 21st century. For the far-future (2070 to 2099), under all climate scenarios the *Swiss Plateau* and the *Downstream Lake* thermal regime stations remain as the stations with the severest extreme events, while the increase in extreme event severity increases the most for the *Regulated* and the *Swiss Plateau* thermal regimes. As the *Swiss Plateau* and *Regulated* thermal regime stations are mostly located in the Swiss low land in the Northwestern part of Switzerland (see Figure 2b), they are the ones that are expected to experience the most severe low flow conditions, especially in summer months under the RCP8.5 scenario, with a discharge reduction ranging from 5 to 60 % (FOEN, 2021; Brunner, et al., 2019; Brunner, et al., 2019; CH2018, 2018). The largest increase from the reference to the far-future period was found at stations for the *Regulated* thermal regime (mean extreme event severity increase RCP2.6: +0.28 °C, RCP4.5: +0.54 °C, RCP8.5: +0.93 °C) followed by the *Swiss Plateau* (RCP2.6: +0.26 °C, RCP4.5: +0.48 °C, RCP8.5: +0.78 °C), *Alpine* (RCP2.6: +0.23 °C, RCP4.5: +0.45 °C, RCP8.5: +0.68 °C), *Downstream Lake* (RCP2.6: +0.23 °C, RCP4.5: +0.40 °C, RCP8.5: +0.61 °C) and *Spring* thermal regimes (RCP2.6: +0.01 °C, RCP4.5: +0.01 °C, RCP8.5: +0.03 °C).

L515-525: these are very general sentences which fit better in an introduction section than in a conclusion

Sentences has been moved to the introduction.  Line 80 to 90

The study of climate change includes the investigation of physical processes on global, regional and local scales. As scales change so too does the required level of detail needed to resolve the different water cycle components that are relevant on the respective scale. An ideally suited approach to address this challenge in hydrological modeling is a multi-fidelity model framework, which combines multiple computational models of varying complexity in an automated selection framework that ensures robust predictions while limiting the computation to only the necessary level of detail (Fernández-Godino, 2023). The use of process dependent fidelity ensures proper representation of physical processes on regional to local scales while keeping computational costs to a minimum. Multi-fidelity modeling is especially useful when acquiring high-accuracy data is costly and/or computationally intensive, as is the case for climate change impact assessment on the hydrological cycle.

L525: why are the water temperature models of "lower fidelity"?

Due to the simplified level of detail. The water temperature models used here are semi-empirical, meaning detailed physical representation has been simplified thus the low fidelity. The inclusion of empirical climate models brings a high-fidelity element to our study. Importantly, the use of low fidelity water temperature models enables us to use the full set of climate models, and not a selection of representative models required while using high fidelity water temperature models (e.g. Michel et al. 2022), in our nationwide study.

L587-593: why is it important to study these hysteresis effects?

A change in hysteresis as described here with elongated loops points to an added warming effect caused by a decrease in the amount of water being heated, i.e. a change in physical function as less water is more easily heated.

L605-614: I would move these lines to the results section, and provide more explanations on the differences between thermal regimes here in the discussion.

Text worked into section 3.3. see above. Section 4.4 on line 657 to 670now reads

The here proposed *extreme event severity index* together with a removal of the climatic trend during each period, allowed us to investigate the change in the baseline of extreme temperature under each thermal regime considered here. The index is independent of past extreme conditions and relate extremes to the time period being investigated. Like for the water temperature warming rates and trends, the severity of temperature extremes was impacted the most by the choice of the climate scenario, similarly so for thermal regimes as a whole and for individual stations. The largest increase of river temperature extremes occurred under the RCP8.5 scenario, followed by the RCP4.5 scenario. Noteworthy is that under the RCP2.6 scenario, extreme event frequency and severity stayed more or less constant throughout the 21$^{st}$ century. As the discharge projections have been directly considered in the employed multi-fidelity modeling approach, the strong increase in extreme event severity for these stations is thus a direct result of the expected increased occurrence of low flow events, while the seasonal near-surface air temperature changes are mostly responsible for an increasing median of river water temperatures.

L634-652: same comment as above, these lines are more for the results section, which would make that section more digestible.

Text moved to section 3.4 line 530 to 548

Textual comments

L93-94: "multi-fidelity modelling" and "from multiple different fidelity levels", I would avoid such repetitions in the same sentence

Now reads on line 109 to 111

In this study a multi-fidelity modeling approach using two semi-empirical surface water temperature models, air2water and air2stream (Toffolon & Piccolroaz, 2015; Piccolroaz et al., 2013), was employed.

L111-114: this is repetition of what is said above

Removed

Caption Table 1: "hydraulic models" or "hydrological models"?

Corrected

L161: "Only stations...", hydrological measuring stations are meant here, I suppose?

Corrected

L170-172: "For situations ...", You lost me in this sentence. Would it be possible to reformulate more clearly?

Section re-written, see above section 2.2

L374: section title: indicate that the thermal thresholds are for fish.

2.9 Thermal thresholds for fish

L421-422: reformulate "at for each station for with ..."

Now on line 454 to 456 reads

The hysteresis class could be determined for each station with future and present river discharge (47 out of 82 stations). For all stations, climate scenarios, and climate models, the index found solutions in hysteresis intervals ranging from 164 to 328 days.

L667: suggestion to just name this section "5. Conclusions"

5. Conclusions

---

## Author Comment (AC2)

We thank the reviewer for constructive comments and provide our answers hereunder.

**RC2**

Råman Vinna and colleagues have used a coupled climate-hydrological-temperature model setup with different levels of representation of reality (fidelity) to simulate river temperatures in Swiss rivers, including future climate projections. They include extreme temperatures and ecological thresholds in their analysis. The topic and scope are rather similar to the Michel et al. (2022, cited in text) paper, also published in *HESS*, but they expand on it by their multi-model approach and additional analysis of thresholds, extremes, and hysteresis. The paper is well-written, clear, and comprehensive, and I did not have major comments, only some minor and mostly technical ones that I outline below.

Minor and technical comments:

- 24 -> "but also in rivers where…"

Corrected.

- 239: -> "inverse stratification"

Corrected.

- 261: I cannot find Table C2. I assume you meant B2?

Yes, it is B2. Corrected.

- 332: "refiling" or "refilling"?

Refilling, corrected.

- Figure 3: Change y-axis to "Water temperature"

Corrected.

- 357: -> "straightforward"

Corrected.

- 362-373: I like this definition of a severity index

Thank you, yes, it is neat to be able to compare across temporal scales and between scenarios.

- 421: "at" or "for"

Corrected.

- 450: -> "from, for example, …"

Corrected.

- The caption of Table 3 mentions yellow marking which does not occur in the table.

It should be *italic*. Table 3, B9 and B10 caption has been updated.

- 525-528: It is unclear from this sentence what vital principle is referred to..

Section reworked, on line 581 to 596 it now reads

4.1 Multi-fidelity modeling approach

The use of semi-empirical models by definition means that some of the physical processes affecting heating is simplified under parameterization and some are directly resolved. The models air2stream and air2water resolve the effect of river depth, discharge, thermal signals from tributaries, inverse stratification in lakes during winter, and seasonal cycles. Parts of the heat balance (e.g. short and longwave radiation) is thus not allowed to change as climate change in our study. However indirectly we consider heat budget changes by using high quality air temperature and discharge projections as input. Glacier retreat is included in the hydrological models providing discharge projections to this study (eg. Muelchi et al., 2021), however for temperature this effect is only indirectly considered in air2stream and air2water through reduced water availability in summer. The effect of high altitude warming as snow and ice recede is not included. Therefore as the cooling caused by melt water recedes, it is expected that warming in high altitude rivers is larger than projected in this study. Yet the lower fidelity water temperature model approach using high-fidelity climate/hydrological model outputs as input enable the important principle of multi-model ensemble, comparison and analysis that is required for robust climate change impact assessments (Duan et al., 2019).

- 552-556: Are there 15 or 16 Alpine stations?

This study includes 16 Alpine stations. Note that station 2462 is not shown in Figures 6 and 7 since Model $M_4$ lacked 30 years of data.

- 585: -> "Rhône"

Corrected.

- 584-586: Could you rewrite this sentence? It is not clear whether lakes or rivers warmed faster in the cited reference.

Paragraph on line 637 to 648 rewritten to:

In terms of overall warming, the strongest warming on an annual basis emerged for stations in the Alpine thermal regime, followed, in order, by stations in the Downstream Lake, Regulated, Swiss Plateau, and Spring thermal regimes (Figure 4). The strong warming of Alpine regime stations has its origins in the strongest near-surface air temperature warming trend in summer that is occurring in southern parts of Switzerland (CH2018, 2018). The strong

warming in the Downstream Lake thermal regime can be explained by the extended residence time of water in lakes compared to rivers in general (allowing longer time for waters to heat up) and to a difference in seasonal patterns, aspects that the employed air2water model explicitly considers. A previous coupled modeling study by the author showed that future lake surface waters (epilimnion) heat faster compared to river waters, with a difference in warming trends between Lake Biel and the Aare River of +0.03 °C per decade and between Lake Geneva and the Rhône River of +0.11 °C per decade (Råman Vinnå et al., 2018).

- 625: "brown trout's" -> "brown trout"

Corrected.

- 675-676: 0.37 °C/decade over 11 decades would be a 4 °C increase. Could you double check the numbers? Moreover, the Results mention a 3.18 °C increase total and 0.36 °C/decade, so please standardise these values.

Corrected. The change is from 2020 to 2099 (8 decades) corrected in manuscript. Line 707 to 710

According to the simulations, for the high emission climate scenario (RCP8.5), average river water temperatures across Switzerland will increase by 3.2 °C (0.36 °C per decade from 2020 to 2099), while under the low emission scenario (RCP2.6) temperatures increase by only 0.9 °C.

- Discussion: One major process for river temperature in the studied systems seems to be the role of disappearing glaciers and snow cover. This is also likely an important factor for changing hysteresis patterns. The discharge models used in the paper may take this into account, but air2water/air2stream do not (e.g. L. 280). It may be valid to assume a nonlinear response of river temperature to disappearing snow/ice that may not be reflected well in the training data (especially for Alpine streams and their low number of stations with long measurement time series). It would be good to add a short paragraph to the Discussion on how this may affect the projections in this paper.

Section 4.1 line 581 to 607 now reads:

The use of semi-empirical models by definition means that some of the physical processes affecting heating is simplified under parameterization and some are directly resolved. The models air2stream and air2water resolve the effect of river depth, discharge, thermal signals from tributaries, inverse stratification in lakes during winter, and seasonal cycles. Parts of the heat balance (e.g. short and longwave radiation) is thus not allowed to change as climate change in our study. However indirectly we consider heat budget changes by using high quality air temperature and discharge projections as input. Glacier retreat is included in the hydrological models providing discharge projections to this study (eg. Muelchi et al., 2021), however for temperature this effect is only indirectly considered in air2stream and air2water through reduced water availability in summer. The effect of high altitude warming as snow and ice recede is not included. Therefore as the cooling caused by melt water recedes, it is expected that warming in high altitude rivers is larger than projected in this study. Yet the lower fidelity water temperature model approach using high-fidelity climate/hydrological model

outputs as input enable the important principle of multi-model ensemble, comparison and analysis that is required for robust climate change impact assessments (Duan et al., 2019).

To expand on previous results of river water temperature projections for Switzerland (Michel et al., 2022), we employed a multi-fidelity modeling approach able to automate the generation of water temperature simulators for the different national river temperature monitoring stations of Switzerland, as summarized in Figure 1. Models of varying complexity were built from integrating high-fidelity climate and hydrological modeling outputs (i.e., downscaled climate (Table 1) and hydrological model outputs (Figure 2a), CH2018 and Hydro-CH2018) with low-fidelity river temperature models of varying degrees of parametrization i.e., air2water and air2stream (Toffolon & Piccolroaz, 2015; Piccolroaz et al., 2013). Statistical learning-based coupling of atmospheric and hydrological stations (Table 2) and classification of river stations into thermal regimes (Figure 2b & 2c) enabled optimal low-fidelity model selection (Figure 2d) and parametrization.

---

## Author Comment (AC3)

We thank the reviewer for constructive comments and provide our answers hereunder.

**RC3**

This paper introduces modelling climate change impacts to river temperatures in Switzerland. To do this, the authors conducted a multi fidelity modelling method which uses statistical pattern recognition to estimate river water temperatures under climate change and thereby close the aforementioned spatial gap by determining, in an automated manner and on a country-wide scale, how future river water temperatures are likely going to change.

The authors frequently refer to their method as novel. I suggest to remove all occurrences of this claim of novelty. Simply describe the model used. Some would argue that the discipline of stream temperature modelling has advanced beyond the use of air temperature and discharge alone for predicting river temperature, regardless of whether it is focused on current versus future climate. While it may be practical for a nationwide attempt, and in that case also 'efficient', it is not necessarily "novel".
The word novel occurred once on line 93, it has been removed from the manuscript.

The reason physically-based models require a lot of data is because they attempt to represent mechanisms and therefore attribute causality of rising river temperatures. River temperatures are a function of many processes beyond simply river discharge and air temperature, as has been discussed in recent literature. The limitation of the "efficient" model approach is that many, many physical drivers of river warming are completely ignored. In predictions of stream temperature, simplifying the "*more complex processes into purely empirical parameters*" often involves using lumped parameters and lumped heat exchange coefficients which ignore aspects of climate change, especially with respect to the shortwave and longwave radiation balance, and increases in atmospheric emissivity which is driving the air temperature warming. There is not a single mention of any of this. The authors simplify the controls to the energy balance as being based on discharge and air temperature, which is not complete, nor does it use best-available-science. If the authors make simplifications in processes, and vary the number of parameters used across their different simulations in order to get a nation-wide dataset for Switzerland, they need to be very clear about this approach and also be upfront about the many, many limitations of their results.

Physical drivers are not ignored in this kind of study, they are included indirectly through parameterization. One can correctly argue that being included under a parameter constitute being frozen in time. Since we use air temperature as input from Ch2018, we capture part of the changes over time in the surface heat budget relevant for water temperature. A more complete heat budget was included for snow and glacier melt in Hydro-CH2018, which provides the discharge to this study.
Apart from the effect of air temperature on water temperature, the models additionally resolve the effect of river discharge, depth, thermal signals from tributaries, inverse stratification in lakes during winter, and seasonal cycles.
The manuscript has been updated in both the introduction, method and discussion sections to make it clearer for the reader the limitations and advantages of our approach.

Line 60 to 107 in section 1

A common challenge for model-based studies is the question of the optimal model to use. In surface hydrological applications, models can broadly be split into two major groups: process-based and statistical/stochastic models (Benyahya et al., 2007). Process-based models are based on physical equations and can resolve many hydrological processes in a physically robust manner, from the local to the catchment scale. However, albeit physically more robust, process-based models generally require a significant amount of input data and computational resources for the simulation of hydrological processes on the catchment scale, therefore limiting their applicability for climate change analyses on national scales. Statistical/stochastic models, as opposed to process-based models, are data driven, that is, are based on empirical relationships between input and output data. While they are physically less robust, their advantage lies in their relative simplicity and limited data requirements, sacrificing detail for increased repeatability and spatial coverage. However, in order to build on the efficiency of statistics whilst preserving a clear physical basis, as a compromise between the two major model groups, a sub-group of semi-empirical models, which employs physically meaningful equations but simplifies the more complex processes into purely empirical parameters, was developed (Piccolroaz et al., 2013). These semi-empirical models are ideally suited for hydrological climate change projections, as they provide much more robust projections compared to purely statistical approaches but simultaneously allow for a more comprehensive analysis than process-based models by enabling multi-model climate change ensemble analyses (La Fuente et al., 2022; Meehl et al., 2007).

The study of climate change includes the investigation of physical processes on global, regional and local scales. As scales change so too does the required level of detail needed to resolve the different water cycle components that are relevant on the respective scale. An ideally suited approach to address this challenge in hydrological modeling is a multi-fidelity model framework, which combines multiple computational models of varying complexity in an automated selection framework that ensures robust predictions while limiting the computation to only the necessary level of detail (Fernández-Godino, 2023). The use of process dependent fidelity ensures proper representation of physical processes on regional to local scales while keeping computational costs to a minimum. Multi-fidelity modeling is especially useful when acquiring high-accuracy data is costly and/or computationally intensive, as is the case for climate change impact assessment on the hydrological cycle.

Given the past and future changes to Swiss river water temperatures and considering both the high sensitivity of aquatic species to river water temperatures and the increasing demand for river water by agriculture, industry and society as a whole, it is critical to obtain a robust spatial and temporal understanding of the temperature increases that are expected for the many different rivers and streams of Switzerland. Here, we developed an efficient multi-fidelity modeling method guided by statistical pattern recognition to estimate river water temperatures under climate change and thereby close the aforementioned spatial gap by determining, in an automated manner and on a national scale, how future river water temperatures are likely going to change. Compared to previous projections of climate warming in Swiss rivers (Michel et al., 2022), the simplified multi-fidelity modeling approach not only enabled to cover the national scale (+90%) but also further thermal regimes (here 5, previously 2) and based on 22 GCM-RCM chains (previously 7). By grouping catchments together via statistical pattern recognition, we were able to classify rivers (including spring-fed rivers) into 5 different thermal regimes, improving model results by allowing for optimal model selection at each station and enabling regime-specific analyses. The effect on warming by changing river discharge was

investigate through a hysteresis analysis. Additionally, we introduce the *extreme event severity* index as an analytic tool to evaluate the change in thermal extreme amplitude.

Line 261 to 273 in section 2,5 Surface water temperature model setup
Both models include up to eight parameters ($a_1$ to $a_8$) which are fitted towards measured data. Apart from the effect of air temperature on water temperature, the models additionally resolve the effect of river depth, discharge, thermal signals from tributaries, inverse stratification in lakes during winter, and seasonal cycles. Model complexity, i.e. how many processes are directly being resolved by the models or indirectly included through parameter estimation, can be varied by removal of one or more of the additional processes listed above, resulting in the use of 8, 7, 6, 5, 4 or 3 parameters. Depending on local conditions, model performance can be improved by the removal of processes which play a minor or insignificant role for water temperature. Where this simplification with removal of parameters was done (Table B2), removed processes plays a minor role for the simulation of water temperature as evident from decreased model performance while being included. For additional information about *air2stream* and *air2water* see Appendix A and Piccolroaz et al. (2013) and Toffolon & Piccolroaz (2015).

Lines 302 to 318, Section 2.6 Trend correction

Empirical models generally predict less warming in the future compared to physically based models, the primary reason being underrepresentation of the thermal catchment memory, including snow and ice (Leach & Moore, 2019). To quantify how good the models air2stream and air2water, which both lack deterministic considerations of snow and ice melt, are able to recreate past trends, we compared trends from river water temperature measurements and corresponding modeled temperature trends between 1990 and 2019. On an annual basis, this comparison was possible for 25 out of 82 river stations, consisting of 9 Downstream Lake, 7 Regulated, 7 Swiss Plateau, 2 Alpine, and 0 Spring thermal regime river stations. Stations were selected with a 30 years of continuous data requirement in air and water temperature and river discharge. Only statistically significant trends ($p < 0.05$) were considered.

Both air2stream and air2water underestimate the annual temperature trend during the reference period on average by 0.14 and 0.11 °C per decade, respectively. For air2stream, the annual trend bias is smallest for the Swiss Plateau thermal regime (0.09 °C per decade) and largest in the Alpine thermal regime (0.17 °C per decade). Seasonally, the trend bias is largest from June to August and September to November, whereas, especially for air2water, the bias is small from December to February and March to May.

Line 580 to 596
4 Discussion
4.1 Multi-fidelity modeling approach
The use of semi-empirical models by definition means that some of the physical processes affecting heating is simplified under parameterization and some are directly resolved. The models air2stream and air2water resolve the effect of river depth, discharge, thermal signals from tributaries, inverse stratification in lakes during winter, and seasonal cycles. Parts of the heat balance (e.g. short and longwave radiation) is thus not allowed to change as climate

change in our study. However indirectly we consider heat budget changes by using high quality air temperature and discharge projections as input. Glacier retreat is included in the hydrological models providing discharge projections to this study (eg. Muelchi et al., 2021), however for temperature this effect is only indirectly considered in air2stream and air2water through reduced water availability in summer. The effect of high altitude warming as snow and ice recede is not included. Therefore as the cooling caused by melt water recedes, it is expected that warming in high altitude rivers is larger than projected in this study. Yet the lower fidelity water temperature model approach using high-fidelity climate/hydrological model outputs as input enable the important principle of multi-model ensemble, comparison and analysis that is required for robust climate change impact assessments (Duan et al., 2019).

*Lines 122-125:* It is unclear how many years of actual data were used. This must be clarified. In one sentence, they say at least 1 year, in another sentence they say *"data should preferably cover 30 years"*. Authors need to specify which simulations used which timespan of datasets, as this is a fundamental influence on the accuracy of the predictions you are reporting in your Results section.

The duration of datasets used for calibration and validation are given in Table B2 and are described in section 2.5. The following section has been moved from section 2.5. to 2.1.
"Temporally overlapping, daily averaged near-surface air temperature and river discharge measurements spanning the 30-year reference period of 1990 to 2020 were used as calibration data, while for validation the data from 1980 to 1990 were used (Table B2 in Appendix). By choosing to use the most recent data for calibration rather than validation ensures that recent local climate conditions are carried into future projections (Shen et al., 2022). For the few cases where no forcing data for calibration did exist between 1990 to 2020 (Table B2), validation was deprioritized and calibration performed for the 1980-1990 data."

Lines 151-153: The authors state, "*For monitoring stations at which historic river discharge data or future river discharge projections weren't available, only future near-surface air temperature projections were used to simulate water temperature.*" This is a major limitation. For how many stations did the authors predict river temperature only from air temperature alone? And how do you correct for the fact that come used discharge and some didn't use discharge, but you are presenting the results of those two different simulation approaches as being equal in your Results section?

Lines 154-156: Many studies have demonstrated that the resolution of the climate model data will influence your results. Here the authors state, "*Where climate projections were available at multiple different spatial resolutions (i.e. 0.11° and 0.44°), only one model, as indicated in Table 1, was included in the analysis, following the approach of Muelchi et al., 2021.*"

These two items above both will affect the model results, potentially significantly. Sometimes the authors use air temperature and discharge to predict river temperature. Sometimes the authors use only air temperature to predict river temperature (many authors have shown this is not sufficient). Sometimes the authors used 0.11° spatial resolution and sometimes they used 0.44° resolution. How are the results defensible and comparable?

We agree with the reviewer that discharge is an important parameter for modeling water temperature and should be used wherever applicable. Here at 47 out of 82 stations we could use river discharge. 35 stations were modeled without discharge (Table B1).

Our study combines a wide variety of datasets (measured and modelled) with varying degree of data availability and accuracy. In the multi-fidelity modeling approach, we do not rank the inputs from climate models or measurements. Nor do we select "representative" model runs or climate scenarios.

Instead, the simplicity of this method enables us to use a wide range of climate models, flow models and water temperature models. Through the use of ensembles and combined analysis inconsistencies and biases included in all data and models are smooth out. This follows recommendations from recent climate model downscaling in Switzerland: "To account for the inherent climate model and greenhouse gas scenario uncertainty, we also advise users to employ a maximum number of CH2018 simulations (CH2018 project team)."

A trend correction was preformed to correct for seen discrepancies between our models and measurements during the reference period. The correction needed was smallest for the *air2water* model (18 stations) compared to *air2stream* (17 stations). The *air2water* model which works completely without discharge outperformed *air2stream* downstream of lakes, this indicates that despite lacking the input of discharge we could model the impact of climate change satisfactionally without river discharge see section 2.6 above.

Lines 165-172: Again, the deviation across methods raise concerns for presenting comparable results. This study employs large datasets which require some level of computational proficiency, but it appears they did not employ spatial interpolation methods of weather data across elevation or across distance. It is very common (and not difficult) to employ spatial interpolation methods of time-series weather data to a particular river location, in order to produce more accurate results at a specific distance along a river. The authors state: "*Meteorological stations were subsequently paired with hydrological stations such that (a) the horizontal distance between river and meteorological stations was minimal (criterion "DIS"), (b) the meteorological station was representative of the conditions in the upstream drainage area (criterion "DRA"), and (c) the elevation difference didn't exceed a reasonable threshold of 200 m (criterion "ELE"). Where possible, all three criteria were adhered to. For situations where the closest meteorological station was either not fulfilling DRA or ELE, the DIS criterion was evaluated only for stations which fulfilled both DRA and ELE.*" While this explanation is, in theory, reproducible, I am not sure that adjusting the criteria on a station-by-station basis is defensible. Authors need to address this.

Both the *air2stream* and *air2water* models use representative atmospheric forcing for a drainage area above a certain point to model water temperature. Simulations are thus conducted towards all relevant heat transfers taking place upstream of this point as captured by water temperature measurements obtained at the point. Thus, for these models the exact location of atmospheric forcing in the drainage area is of minor importance. It is far more important to have representative meteorological conditions, hence the selection criteria's above. Anny remaining inconsistency between the actual dataset used as input and how local atmospheric conditions affect water temperature in the drainage area, are compensated for in the calibration of the two models with up to 8 parameters. By preforming spatial interpolation of meteorological data and climate model results unknown biases are created, especially in

settings with pronounced relief (the Alps), bias which increase with the distance from each station.
In Ch2018, regional climate models were downscaled with quantile mapping towards measurements at local stations. Naturally, the quality of this downscaling improved towards the meteorological stations with minimal climate model bias right next to each station. Thus, by selecting to use the downscaled climate model data delivered at the location of the meteorological stations, climate model bias was minimized.
For processes such as stream discharge, spatial and temporal distribution of precipitation and snow/glacier melt is more important compared to heat budget processes for water temperature modeling. Spatial dependency was considered in Hydro-Ch2018, those discharge projections we use here.

Lines 322-324: What do the authors mean by "shape-preserving interpolation" across multiple days without data, and where is this interpolation method presented in this paper? Authors state: "*Before adjusting the water temperature model output from 1990 to 2099, Bcs was combined into a continuous dataset by filling in the 3- to 5-day gap in between each season with shape-preserving interpolation.*"
Now reads on lines 348to 352:
"Before adjusting the water temperature model output from 1990 to 2099, the seasonal $Bc_s$ was combined into a continuous dataset $Bc$. To avoid a sharp shift in $Bc$ between each season, a 3- to 5-day gap in between each season was smoothed with shape-preserving interpolation (Piecewise cubic Hermite interpolation, PCHIP; MATLAB© R2022a)."

Line 439: "*Considering only the far future*" ◊ what do the authors mean by "far future". Please clarify.
Far-future period (2070 to 2099), is defined in the manuscript.

The authors' most significant result is summarized by "*Climate change impact was heterogeneous between stations, yet common patterns were found within thermal regimes*". It is concerning to present results when each result was achieved through a subtle deviation from the methods, the spatial resolution of inputs, the handling of missing days of data, and even using different model inputs. In some simulations the only model input is air temperature. How can results and hysteresis loops be viewed as comparable across simulations by the reader, when the methods employed to get there were modified, changed, required deviation of some methods, used a different number of parameters in 'air2water'/'air2stream' (i.e. Line 695 "*adapting their parametrization complexity to the required level*"), or were slightly different methods across simulations?

The reviewer is correct that the methods differed for each station. However, this was an intended and needed but not a random process. The process, which is known as multi-fidelity modelling, selects for each station the best possible model according to the available data (in this context meaning the model with optimal model complexity as warranted by the data). It would of course be desirable to have an identical data basis for all stations, but this is the real-world, and in the real-world, this is absolutely never the case. Hence, in order to project river water temperatures for real-world measurement stations, one is left with the choice to either use the lowest complexity model for all stations, as warranted by the station with the poorest data basis for projection, which would lead to comparable but underwhelming projections, or

one can choose the optimal model complexity as warranted by the data availability of each individual station, producing, for all stations, projections with the highest fidelity. We chose, in agreement with the multi-fidelity modelling approach and philosophy, to compare the projections of all stations based on their highest fidelity model and data basis. This is the most appropriate approach to compare and judge projections for real-world stations. For more precise viewpoints we referee to our previous answers in this review.

---

## Author Response (AR2)

**Report #1**

**Our replies to the last comments in red.**

Submitted on 12 Aug 2025

Anonymous referee #2

Anonymous during peer-review: Yes No

Anonymous in acknowledgements of published article: Yes No

Checklist for reviewers

**1) Scientific Significance**

Does the manuscript represent a substantial contribution to scientific progress within the scope of this journal (substantial new concepts, ideas, methods, or data)?

Excellent Good Fair Poor

**2) Scientific Quality**

Are the scientific approach and applied methods valid? Are the results discussed in an appropriate and balanced way (consideration of related work, including appropriate references)?

Excellent Good Fair Poor

**3) Presentation Quality**

Are the scientific results and conclusions presented in a clear, concise, and well structured way (number and quality of figures/tables, appropriate use of English language)?

**Excellent** Good Fair Poor

For final publication, the manuscript should be

accepted as is

**accepted subject to technical corrections**

accepted subject to minor revisions

reconsidered after major revisions

rejected

Were a revised manuscript to be sent for another round of reviews:

**I would be willing to review the revised manuscript.**

I would not be willing to review the revised manuscript.

Suggestions for revision or reasons for rejection

(visible to the public if the article is accepted and published)

The authors replied adequately to most of my comments and I attach a few more comments related to my previous input or the new text, but these should not hinder eventual publication.

**Comments:**

- My comment on L. 24 was not corrected. Please change to "but also in rivers where..."

**Corrected**

- L. 499-500: double use of "from"

**One deleted**

- L. 601-604: These two sentences are a bit unclear and I'm not sure if this in line with the models used. I am not an expert in these models, but as far as I understand, air2stream/water do not explicitly simulate individual heat flux components, but estimate a total heat flux based on air and water temperature (modified by factors such as those described in the sentence before this one). Sentence 603-604 would therefore be a good argument for reliable future predictions, IF this air-water temperature relation is (roughly) maintained in a future climate. If the relative contributions of the heat budget change strongly, this might not be the case; is this what you are trying to say? Please see if you can clarify these sentences.

**Section now reads:**

The use of semi-empirical models by definition means that some of the physical processes affecting heating are simplified under parameterization and some are directly resolved. The models air2stream and air2water resolve the effect of river depth, discharge, thermal signals from tributaries, inverse stratification in lakes during winter, and seasonal cycles. The heat flux between the atmosphere and surface waters (latent and sensible heat, short and longwave radiation) is not directly resolved by air2stram and air2water. However, indirectly we consider climate related heat budget changes in our method, through the use of high-quality projections of air temperature and discharge as model input. Glacier retreat is included in the hydrological models providing discharge projections to this study (eg. Muelchi et al., 2021), however for temperature this effect is only indirectly considered in air2stream through reduced water availability in summer. The cooling effect on river water caused by meltwater from snow and ice does not change in our method, as snow and ice recede in a future climate it is expected that warming in high altitude rivers is larger than projected in this study. Therefore, if the relationships between discharge and air temperature towards water temperature remain similar in the future, our method can be used to reliably project future river temperatures. Importantly, the lower fidelity water temperature model approach used here combined with high-fidelity climate/hydrological model outputs as input enable the principle of multi-model ensemble, comparison and analysis that is required for robust climate change impact assessments (Duan et al., 2019).

- L. 636-639: The authors still switch between 15 and 16 Alpine stations; in their reply, you mention that station 2462 was left out of some figures, but in these sentences, the difference is still confusing. Please shortly clarify in the text.

It is 16 Alpine Stations. Manuscript has been corrected and notation for Figure 4, 6, 7 improved to enhance clarity.